

# Surrogate-assisted Bayesian inversion for landscape and basin evolution models

Rohitash Chandra[1,2], Danial Azam[1], Arpit Kapoor[3], and R. Dietmar Müller[1]

[1]EarthByte Group, School of Geosciences, University of Sydney, NSW 2006, Sydney, Australia
[2]Centre for Translational Data Science, University of Sydney, NSW 2006, Sydney, Australia
[3] Department of Computer Science and Engineering, SRM Institute of Science and Technology, Tamil Nadu, India

**Correspondence:** Rohitash Chandra (rohitash.chandra@sydney.edu.au)

**Abstract.**

The complex and computationally expensive features of the forward landscape and sedimentary basin evolution models pose a major challenge in the development of efficient inference and optimization methods. Bayesian inference provides a methodology for estimation and uncertainty quantification of free model parameters. In our previous work, parallel tempering

Bayeslands was developed as a framework for parameter estimation and uncertainty quantification for the landscape and basin evolution modelling software Badlands (Chandra et al., 2018c). Parallel tempering Bayeslands features high-performance computing with dozens of processing cores running in parallel to enhance computational efficiency. Although parallel computing is used, the procedure remains computationally challenging since thousands of samples need to be drawn and evaluated. In large-scale landscape and basin evolution problems, a single model evaluation can take from several minutes to hours, and in certain

cases, even days. Surrogate-assisted optimization has been with successfully applied to a number of engineering problems (Ong et al., 2003; Zhou et al., 2007). This motivates its use in optimisation and inference methods suited for complex models in geology and geophysics. Surrogates can speed up parallel tempering Bayeslands by developing computationally inexpensive surrogates to mimic expensive models. In this paper, we present an application of surrogate-assisted parallel tempering where that surrogate mimics a landscape evolution model including erosion, sediment transport and deposition, by estimating

the likelihood function that is given by the model. We employ a machine learning model as a surrogate that learns from the samples generated by the parallel tempering algorithm and the likelihood from the model. The entire framework is developed in a parallel computing infrastructure to take advantage of parallelization. The results show that the proposed methodology is effective in lowering the overall computational cost significantly while retaining the quality of solutions.

# 1 Introduction

The Bayesian methodology provides a probabilistic approach for the estimation of free parameters in complex models (Sambridge, 1999; Neal, 1996). Hence, a deterministic geophysical forward model can be seen as a probabilistic model via Bayesian





inference which provides a rigorous approach to uncertainty quantification as opposed to optimization methods. The approach is also known as Bayesian inversion which has been used for landscape evolution (Chandra et al., 2018c, a), geological reef evolution models (Pall et al., 2018) and other geo-scientific models (Sambridge, 1999, 2013). A Markov Chain Monte Carlo method (MCMC) can be used to implement Bayesian inference for estimation and uncertainty quantification of free param-

eters (Hastings, 1970; Metropolis et al., 1953; Neal, 2012, 1996). Parallel tempering is a MCMC method that (Marinari and Parisi, 1992; Geyer and Thompson, 1995) features multiple replicas to provide global and local exploration which makes them suitable for irregular and multi-modal distributions (Patriksson and van der Spoel, 2008; Hukushima and Nemoto, 1996). In contrast to canonical MCMC sampling methods, parallel tempering can be more easily implemented in a multi-core or parallel computing architecture (Lamport, 1986).

In our previous work, parallel tempering Bayeslands was presented as a framework for parameter estimation and uncertainty quantification for Badlands, a landscape and basin evolution evolution software (Chandra et al., 2018c). Parallel tempering Bayeslands features high performances parallel computing to enhance the efficiency of estimating free parameters of a Badlands model. Although parallel computing is used, the procedure remains computationally challenging since thousands of samples need to be drawn and evaluated (Chandra et al., 2018c). In large scale landscape evolution problems, a single model

can take hours to days. Hence, it is useful to find ways to improve parallel tempering Bayeslands. Such problems are common for complex forward models of physical processes that can take several hours to days and months to evaluate a single model run. One of the ways to address this problem is using surrogate-assisted estimation.

Surrogate assistant optimization refer to the use of statistical and machine learning models to built approximate simulation of the actual model (Jin, 2011). Many optimization methods lack a rigorous approach for uncertainty quantification, leading

to Bayesian inversion as an alternative, particularly for complex geophysical numerical models (Sambridge, 2013, 1999). The major advantage of a surrogate model is its computational efficiency when compared to the equivalent numerical physical forward model (Ong et al., 2003; Zhou et al., 2007). In the optimization literature, surrogate usage is also known as response surface methodologies (Montgomery and Vernon M. Bettencourt, 1977; Letsinger et al., 1996) that have been applicable for a wide range of engineering problems (Tandjiria et al., 2000; Ong et al., 2005) such as aerodynamic wing design (Ong et al.,

2003). A number of approaches have been used to improve the way surrogates are utilized. (Zhou et al., 2007) combined global and local surrogate models to accelerate evolutionary optimization. (Lim et al., 2010) presented a generalized surrogate-assisted evolutionary computation framework to unify diverse surrogate models during optimization and taking into account uncertainty in estimation. Jin (Jin, 2011) reviewed a range of problems such as single, multi-objective, dynamic, constrained, and multi-modal optimization problems (Díaz-Manríquez et al., 2016). In the Earth sciences, examples for surrogate assisted

approaches include modeling water resources (Razavi et al., 2012; Asher et al., 2015), atmospheric general circulation models (Scher, 2018), computational oceanography (van der Merwe et al., 2007), carbon-dioxide ($CO_2$) storage and oil recovery (Ampomah et al., 2017) and debris flow models (Navarro et al., 2018).

Given that Bayesian inversion is implemented using parallel tempering, parallel computing infrastructure is required and the challenge is how to incorporate surrogates across different processing cores. Recently, surrogate-assisted parallel tempering

has been developed for Bayesian neural networks which presents a global-local surrogate framework where surrogate training



is executed in the master processing core that is used to manage the replicas running in parallel (Chandra et al., 2018b). The method gives promising results, retaining classification accuracy while lowering computational time.

In this paper, we present an application of surrogate-assisted parallel tempering (Chandra et al., 2018b) for Bayesian inversion of surface process models that employ parallel computing infrastructure. We use the Badlands landscape evolution model

(Salles and Hardiman, 2016) as a case study to demonstrate the framework. Overall, the framework features the surrogate-model which mimics the Badlands model and estimates the likelihood function to evaluate the proposed parameters. We employ a neural network model as the surrogate that learns from the history of samples proposed by the parallel tempering algorithm. The entire framework is developed in a parallel computing infrastructure to take advantage of parallelism for surrogate-assisted parallel tempering. We apply the method to several selected benchmark landscape evolution and sediment transport/deposition

problems and show the quality of the estimation of the likelihood given by the surrogate when compared to the actual Badlands model.

## 2   Background and Related Work

### 2.1   Bayesian inference via Parallel tempering

Bayesian inference is based on Bayes theorem and typically implemented by employing MCMC sampling methods to update

the probability for a hypothesis as more information becomes available. The hypothesis is given by a prior probability distribution (also known as the prior) that expresses one's belief about a quantity (or free parameter in a model) before some data is taken into account. Therefore, MCMC methods provide a probabilistic approach for estimation of free parameters in a wide range of models (Raftery and Lewis, 1996; van Ravenzwaaij et al., 2016). The likelihood function is a function of the parameters of a given model provided specific observed data which in the case of landscape evolution would be the ground-

truth topography or the observed sedimentary record. Hence, the likelihood function can be seen as a fitness measure of the proposals. In order to evaluate the likelihood function, one would need to run the given model which in our case is the Badlands model. The likelihood function is used with the Metropolis-criteria to either accept or reject a proposal. When accepted, the proposal becomes part of the posterior distribution which essentially provides the estimation of the free parameter with uncertainties. The sampling process is iterative and requires thousands of samples to be drawn until convergence is reached.

In our case, convergence is defined by a predefined number of samples or until the likelihood function has reached a specific value. Convergence essentially means that the posterior distribution of the given parameters generate Badlands model outputs that resemble ground-truth data (Chandra et al., 2018c).

As noted earlier, parallel tempering is a MCMC method that features parallelism with enhanced exploration capabilities. It features a number of replicas with slight variations in the acceptance criteria through relaxation of the likelihood with a

temperature ladder that affects the acceptance criterion. The replicas associated with higher temperature levels have more chance in accepting weaker proposals (solutions) which could help in escaping a local minimum. Given an ensemble of $N$





replicas defined by the temperature ladder, the state of the ensemble is specified by $X = x_1, x_2, ..., x_N$, where $x_i$ is the replica at temperature level $T_i$. The equilibrium distribution of the ensemble, $X$ is given by

$$\Pi(X) = \prod_{i=1}^{N} \frac{exp(-\frac{1}{T_i} E(x_i))}{Z(T_i)} \tag{1}$$

where $E(x_i)$ is the energy function and $Z(T_i) = \int exp(-\frac{1}{T_i} E(x_i)) dx_i$ is the partition function of the replica at $T_i$. A
Markov chain is constructed to sample $E(x_i)$ at each temperature level $T_i$. At every iteration, the Markov chains can feature two types of transitions that include 1) the Metropolis transition and 2) a replica transition.

In the *Metropolis transition* phase, each replica is sampled independently to perform local Monte Carlo moves defined by the temperature which is implemented by a change in the energy function, $E(x_i)$ for each temperature level $T_i$. The configuration $x_i^*$ is sampled from a proposal distribution $q_i(.|x_i)$ and the Metropolis-Hastings ratio at temperature level $T_i$ is given as

$$L_{local}(x_i \rightarrow x_i^*) = exp(-\frac{1}{T_i}(E(x_i^*) - E(x_i))) \tag{2}$$

where, $L$, represents the likelihood at the *local* replica and the new state is accepted with probability $min(1, W_L(x_i \rightarrow x_i^*))$. The detailed balance condition holds for each replica and therefore it holds for the ensemble system.

The *Replica transition* phase considers the exchange of current state between two neighbouring replicas based on the Metropolis-Hasting acceptance criteria. Hence, given a probability $\alpha$, pairs of replica defined by two neighboring tempera-
ture levels, $i$ and $i + 1$ are exchanged.

$$x_i \leftrightarrow x_{i+1} \tag{3}$$

The exchange of neighboring replicas that provide an efficient balance between local and global exploration (Sambridge, 2013). The temperature ladder and replica exchange have been of focus of investigation in the past (Calvo, 2005; Liu et al., 2005; Bittner et al., 2008; Patriksson and van der Spoel, 2008). There is a consensus that they need to be tailored for different
types of problems given by their likelihood landscape. In this paper, the selection of temperature spacing between the replicas is carried out using a Geometric spacing methodology (Vousden et al., 2015).

$$T_i = T_{max}^{(i-1)/(M-1)} \tag{4}$$

where $i = 1, \ldots, M$ and $T_{max}$ is maximum temperature which is user defined and dependent on the problem.

## 2.2 Badlands and Bayeslands

Landscape evolution models incorporate different driving forces such as tectonics or climate variability (Whipple and Tucker, 2002; Tucker and Hancock, 2010; Salles and Duclaux, 2015; Campforts et al., 2017; Adams et al., 2017) and combine empirical





data and conceptual methods into a set of mathematical equations. *Badlands* (basin and landscape dynamics) (Salles and Hardiman, 2016) is an example of such a model that can be used to reconstruct landscape evolution and associated sediment fluxes (Howard et al., 1994; Hobley et al., 2011). We use *Badlands* (Salles and Hardiman, 2016; Salles, 2016; Salles et al., 2017) to simulate landscape evolution and sediment transport/deposition of selected areas in order to provide estimation with

uncertainty quantification of the free parameters such as precipitation and erodibility.

The Badlands model simulates landscape dynamics which requires an initial topography that is exposed to climate and geological factors over time. In order to create test problems for Badlands, a set of climate and geological parameters defined by $\theta$ needs to be predefined to determine landscape evolution over a given timescale $T$. The final (ground-truth) topography at time $T$ and expected sediment deposits at selected intervals in time are used to evaluate the quality of proposals during

sampling in Bayeslands (Chandra et al., 2018c). Bayeslands features parallel tempering for the estimation of free parameters and uncertainty quantification in model outputs for landscape simulation (Chandra et al., 2018c). Bayeslands estimates a set of free parameters given by $\theta$ that is constrained by some data $\mathbf{D}$. Bayeslands samples the posterior distribution $p(\theta|\mathbf{D})$ using principles of Bayes rule

$$p(\theta|\mathbf{D}) = \frac{p(\mathbf{D}|\theta)p(\theta)}{P(\mathbf{D})}$$

where, $p(\mathbf{D}|\theta)$ is the likelihood of the data given the parameters, $p(\theta)$ is the prior, and $p(\mathbf{D})$ is a normalizing constant and equal to $\int p(\mathbf{D}|\theta)p(\theta)d\theta$. $\theta$ represents the set free parameters such as precipitation and erodibility in the Badlands model and the data $\mathbf{D}$ represents the real topography. The prior distribution (also known as prior) refers to one's belief in the distribution of the parameter without taking into account the evidence or data. The prior distribution is adjusted by sampling from the posterior with given likelihood function that takes into account the data and the model. The goal of Bayeslands is to find estimate the $\theta$

given the posterior distribution such that the simulated topography by Badlands can match the real topography $\mathbf{D}$.

## 3   Methodology

### 3.1   Benchmark landscape evolution problems

We select two benchmark landscape problems presented in parallel tempering Bayeslands (Chandra et al., 2018c) that were adapted from earlier work (Chandra et al., 2018a). These include *Continental Margin (CM)* and *Synthetic-Mountain* landscape

evolution problems which have been chosen due to their computational time required for running a single model. Both of these problems use less than ten seconds to run a single model on a single central processing unit (CPU). These problems are well suited for a parameter evaluation for the proposed surrogate-assisted Bayesian inversion framework. In order to demonstrate an application which is computationally expensive, we introduce another problem, which features the landscape evolution of Tasmania, Australia, for a million years that features the region shown in Figure 1. The Synthetic-Mountain landscape

evolution is a synthetic problem while the Continental-Margin problem is a real-world problem based on the topography of a region along the eastern margin of the South Island of New Zealand as shown in Figure 1. We then use Badlands to evolve the

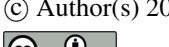


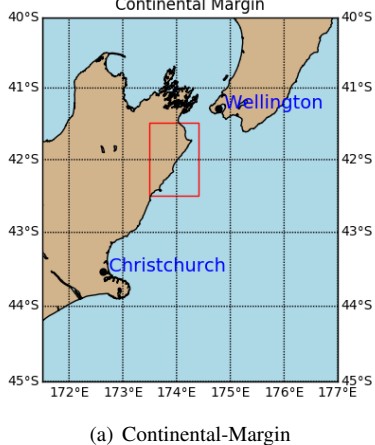
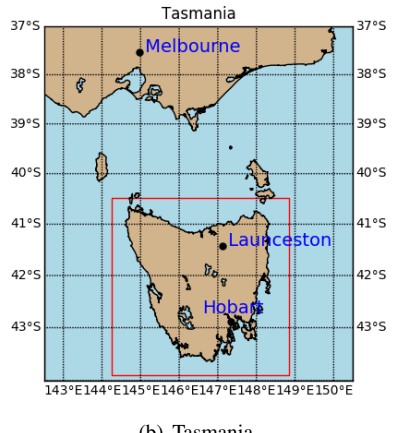

(a) Continental-Margin          (b) Tasmania

**Figure 1.** Location of (a) Continental-Margin problem shown taken from South Island of New Zealand (llcrnrlon =173.5 ° East, llcrnrlat=-42.5° South, urcrnrlon=174.5° East, urcrnrlat=-41.5° South). (b) Tasmania, Australia (llcrnrlon =144.5 ° East,llcrnrlat=-43.5° South, urcrnrlon=148.5° East,urcrnrlat=-40.5° South). Note the following abbreviations: llcrnrlon refers to longitude of lower left hand corner, llcrnrlat refers to latitude of lower left hand corner. urcrnrlon refers to longitude of upper right hand corner and urcrnrlat refers to latitude of upper right hand corner of the desired map domain (degrees).

initial landscape with parameter settings such as rainfall and erodibility given in Table 1 and Table 2 and create the respective problems synthetic ground-truth topography.

The initial and synthetic ground-truth topographies along with erosion-deposition that shows sediment formation for these problems appear in Figure 2 and 3, respectively. Note that the figure shows that the Synthetic-Mountain is flat in the beginning, then given constant uplift rate along with weathering with constant rainfall parameter value, a mountain is formed. We note that we use present-day topography as the initial topography in the Continental-Margin and Tasmania problems, whereas, a synthetic flat region is used as Synthetic-Mountain initial topography. Each of these problems involve an erosion-deposition model history that is used to generate synthetic ground-truth data for the final model state that we then attempt to recover. Hence, the likelihood function given in the following subsection takes both the landscape topography and erosion-deposition ground-truth into account. The Continental-Margin and Tasmania cases feature six free parameters (Table 2) whereas the Synthetic-Mountain features 5 free parameters. Note that the marine diffusion coefficients are absent for the Synthetic-Mountain problem since the region does not cover or overlap with coastal and marine areas. The main reason behind choosing the two benchmark problems is due to their nature, i.e the Synthetic-Mountain problem features uplift rate which is not featured in the Continental-Margin problem. The Continental-Margin problem features other parameters such as the marine coefficients. The Tasmania problem features a much bigger region hence more computational time is used for running a single model. The common feature in all three problems is that they model both the topography erosion-deposition to show sediment formation over time. Furthermore, the priors were drawn from a uniform distribution with lower and upper limit given in Table 3.





(a) Synthetic-Mountain initial topography

(b) Continental-Margin initial topography

(c) Synthetic-Mountain ground truth topography

(d) Continental-Margin synthetic ground-truth topography

(e) Synthetic-Mountain erosion-deposition map

(f) Continental-Margin erosion-deposition map

**Figure 2.** Synthetic-Mountain: Initial and eroded ground-truth topography after a million years of evolution. Continental Margin (CM) : Initial and eroded ground-truth topography and sediment after one million years. The erosion-deposition that forms sediment deposition after one million years is also shown.



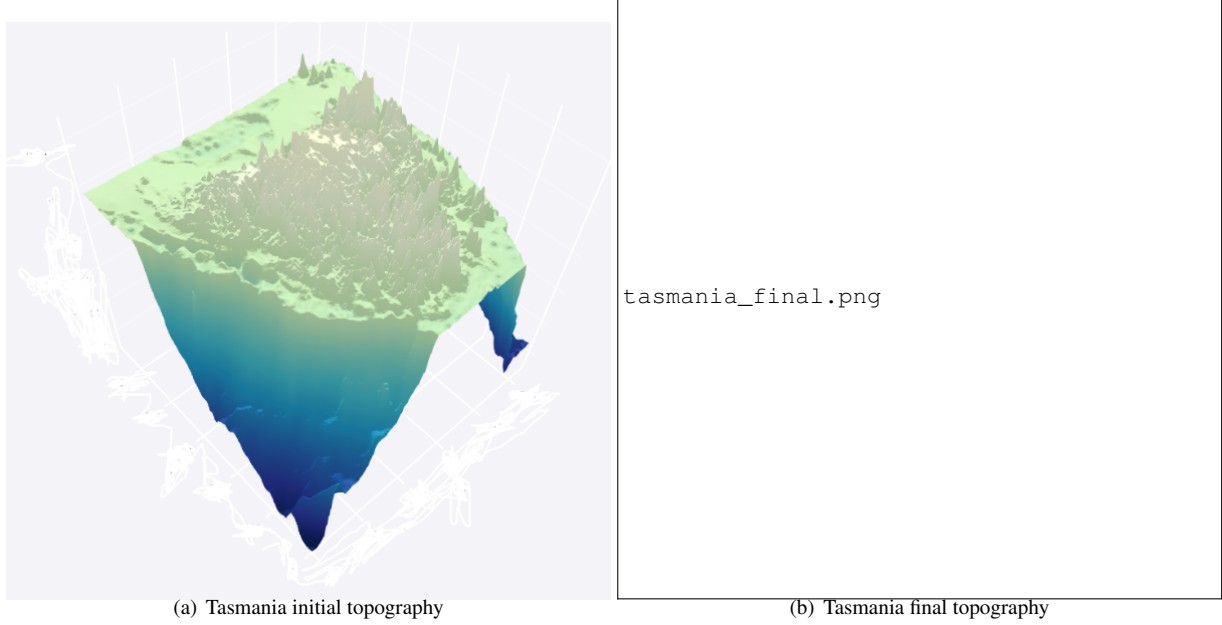

(a) Tasmania initial topography     (b) Tasmania final topography

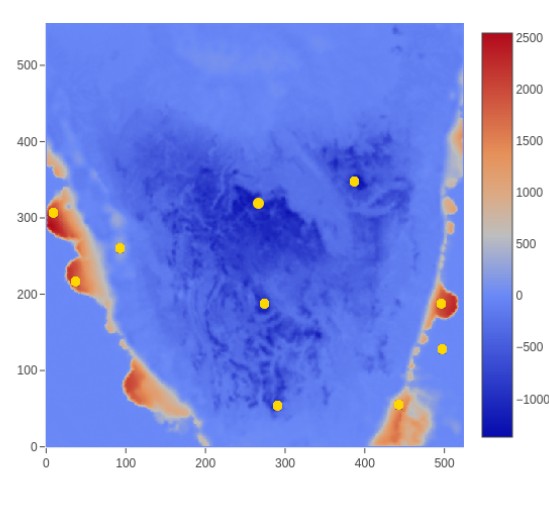

(c) Tasmania erosion-deposition map

**Figure 3.** Tasmania: Initial and eroded ground-truth topography along with erosion-deposition that shows sediment deposition after one million years evolution.





| Topography | Evo.(years) | Length [km, pts] | Width [km, pts] | Res. factor | Run-time (s) |
|---|---|---|---|---|---|
| Continental-Margin | 1 000 000 | [136.0, 136] | [123.0, 123] | 1 | 7.5 |
| Synthetic-Mountain | 1 000 000 | [80,202] | [40,102] | 1 | 10 |
| Tasmania | 1 000 000 | [510,523] | [537,554] | 1 | 71.3 |

**Table 1.** In the given landscape evolution problems, the run-time represents approximately the duration for one model to run on a single central processing unit (CPU).

| Topography | Rainfall (m/a) | Erod. | n-value | m-value | c-marine | c-surface | Uplift (mm/a) |
|---|---|---|---|---|---|---|---|
| Continental-Margin | 1.5 | 5.0-e06 | 1.0 | 0.5 | 0.5 | 0.8 | - |
| Synthetic-Mountain | 1.5 | 5.0-e06 | 1.0 | 0.5 | - | - | 1.0 |
| Tasmania | 1.5 | 5.0-e06 | 1.0 | 0.5 | 0.5 | 0.8 | - |

**Table 2.** True values of parameters

| Topography | Rainfall (m/a) | Erod. | n-value | m-value | c-marine | c-surface | uplift |
|---|---|---|---|---|---|---|---|
| CM-ext. | [0,3.0 ] | [3.0-e06, 7.0-e06] | [0, 2.0] | [0, 2.0] | [0.3, 0.7] | [0.6, 1.0] | - |
| Synthetic-Mountain | [0,3.0 ] | [3.0-e06, 7.0-e06] | [0, 2.0] | [0, 2.0] | - | - | [0.1, 1.7] |
| Tasmania | [0,3.0 ] | [3.0-e06, 7.0-e06] | [0, 2.0] | [0, 2.0] | [0.3, 0.7] | [0.6, 1.0] | - |

**Table 3.** Prior distribution range of model parameters

| Topography | Pt. 1 | Pt. 2 | Pt. 3 | Pt. 4 | Pt. 5 | Pt. 6 | Pt. 7 | Pt. 8 | Pt. 9 | Pt. 10 |
|---|---|---|---|---|---|---|---|---|---|---|
| Continental Margin | (4,40) | (6,20) | (14,66) | (39,8) | (40,5) | (42,10) | (59,13) | (68,40) | (72,44) | (75,51) |
| Synthetic-Mountain | (5,5) | (10,10) | (20,20) | (30,30) | (40,40) | (50,50) | (25,25) | (37,30) | (44,27) | (46,10) |
| Tasmania | (260,320) | (400,350) | (270,180) | (290,50) | (500,120) | (500,195) | (44,200) | (5,315) | (450,50) | (95,260) |

**Table 4.** Erosion-deposition (sediment) coordinates used in likelihood evaluation

## 3.2 Bayeslands Model

The likelihood function captures the quality of topography simulation along with quality of successive erosion-deposition which denotes the sediment thickness evolution through time. This makes the sampling problem more challenging but useful for certain applications. More specifically, the likelihood function evaluates the quality of the proposals by taking into account the difference between the final simulated Badlands topography and the ground-truth topography. The likelihood function also considers the difference between the simulated and ground-truth sediment thickness at selected time intervals. Hence, we use the likelihood function from (Chandra et al., 2018c) which is given as follows.

Let the initial topography be denoted by $\boldsymbol{D}_0$, with $\boldsymbol{D}_0 = (D_{0,s_1} \ldots, D_{0,s_n})$, where $s_i$ corresponds to site $s_i$, with co-ordinates latitude, $u_i$, and longitude, $v_i$. Suppose that we are interested in the topography $t_{max}$ years into the future, we will denote this





by $\boldsymbol{D}_t$, with $\boldsymbol{D}_T$ defined to be the current topography. Hence, the model that generates the process is given by

$$D_{t,s_i} = f_{t,s_i}(\boldsymbol{\theta}) + \epsilon_{t,s_i} \text{ with } \epsilon_{t,s_i} \sim (0,\tau^2) \tag{5}$$

for $t = 0, 1, \ldots, T$ and $i = 1, \ldots, n$, where $\boldsymbol{\theta}$ are the parameters of the Badlands model and $f_{t,s_i}(\boldsymbol{\theta})$ is the output of the Badlands forward model. This essentially states that the topography is function of the Badlands forward model given parameters $\boldsymbol{\theta}$, plus

some Gaussian noise with zero mean and variance $\tau^2$. The likelihood function $L_e(\boldsymbol{\theta})$ is given by

$$L_l(\boldsymbol{\theta}) = \frac{1}{(2\pi\tau^2)^{n/2}} \exp\left\{ -\frac{1}{2} \frac{\sum_{t=1}^T \sum_{i=1}^n (D_{t,s_i} - f_{t,s_i}(\boldsymbol{\theta}))^2}{\tau^2} \right\}$$

where the subscript $e$, in $L_e(\theta)$, denotes the elevation likelihood.

We note that Badlands produces successive time-dependent topographies; however, only the final topography $\mathbf{D}_T$ is used for the calculation of the elevation likelihood, because usually little ground-truth information is available for the detailed evolution

of surface topography. In contrast, sediments preserve the stratigraphic record of the time-dependence of sedimentation and can be used to ground-truth the time-dependent evolution of surface process models that include sediment transport and deposition, as is the case for Badlands. We therefore define another random variable $\mathbf{z}_t = (z_{t,s_1} \ldots, z_{t,s_m})$ which represents the sedimentary record at sites $s_1, \ldots, s_m$. We assume that observed values of $\mathbf{z}_t$ are a function of the Badlands ground-truth forward model, with parameter $\boldsymbol{\theta}$ and some Gaussian noise

$$z_{t,s_i} = g_{t,s_i}(\boldsymbol{\theta}) + \eta_{t,s_i} \text{ with } \eta_{t,s_i} \sim (0,\sigma^2) \tag{6}$$

then the sediment likelihood, $L_s(\theta)$ is

$$L_s(\boldsymbol{\theta}) = \frac{1}{(2\pi\sigma^2)^{mT/2}} \exp\left\{ -\frac{1}{2} \sum_{t=1}^T \sum_{j=1}^m \frac{(Z_{t,s_j} - g_{t,s_j}(\boldsymbol{\theta}))^2}{\sigma^2} \right\}$$

giving a total likelihood $L(\boldsymbol{\theta})$:

$$L(\boldsymbol{\theta}) = L_s(\boldsymbol{\theta}) \times L_l(\boldsymbol{\theta}).$$

The complete set of unknowns in the model is given by $\boldsymbol{\theta}$, $\tau^2$ and $\sigma^2$.



---

**Alg. 1** Surrogate-assisted parallel tempering for Geo-scientific models

**Data**: Actual topography dataset

**Result**: Posterior distribution of free parameters $\theta$ (eg. rainfall and erodibility )

1. Define and initialize $M$ replica $\theta_1$, $\theta_2$, ..., $\theta_M$ with corresponding temperature values $T_1$, $T_2$, ..., $T_M$. 2. Initialize number of samples for replica, $i = 0$. 3. Set replica swap-interval, $R_{swap}$. 4. Set maximum samples for each replica, $R_{max}$. 5. Set surrogate interval, $S_{interval}$. 6. Set surrogate probability, $S_{prob}$. 7. Set maximum samples for entire framework, $F_{max}$. 8. Set number of replicas, $R_{num}$.

**while** *convergence is reached* $(i < F_{max}) / R_{num}$ **do**

**for** *each replica R* **do**

\*Metropolis Transition

      **for** *each i in* $R_{max}$ **do**

Sample $\theta_i^*$ using random-walk as $\theta_i^* = \theta_i + \epsilon$

        Draw $\kappa$ from a Uniform distribution [0,1]

        **if** $\kappa < S_{prob}$ *and* $i > S_{interval}$ **then**

6          Estimate $L_{local}$ from local surrogate's prediction, $L_{surrogate}$

          1. Load global surrogate model parameters to local surrogate

          2. Predict $L_{surrogate}$ value with the proposed $\theta_i^*$.

          3. $L_{past} = \text{mean}(L_{i-1}, L_{i-1}, L_{i-2})$

          4. Assign $L_{local} = (0.5 * L_{surrogate}) + 0.5 * L_{past}$

          5. Save $L_i = L_{local}$

8        **else**

9          $L_{local}$ is calculated using true likelihood function using geo-scientific model (Badlands)

          Draw $\alpha$ from a Uniform distribution [0,1]

          **if** $\alpha \leq L_{local}(\theta_i \rightarrow \theta_i^*)$ **then**

Update chain, $\theta_i \leftarrow \theta_i^*$

**end**

**end**

\* Replica Transition

      **if** $i \bmod R_{swap}$ **then**

Signal master-process to calculate replica transition probability $P(\theta_i \leftrightarrow \theta_{i+1})$

        **for** *each replica* **do**

Draw $\beta$ from a Uniform distribution [0,1]

          **if** $\beta \leq P(\theta_i \leftrightarrow \theta_{i+1})$ **then**

Exchange neighboring Replica, $\theta_i \leftrightarrow \theta_{i+1}$

**end**

**end**

**end**

**if** $i > R_{max}$ **then**

\*Adapt: $T_{replica} = 1$

        (Move to canonical MCMC for local exploration)

**end**

**if** $i \bmod S_{interval}$ **then**

Signal master process

        Set $\Theta$ which features history of proposals $\Phi(\theta)$ and response $\lambda(L_{local})$

**end**

**end**

\*Global Surrogate Training

    **if** $t \bmod S_{interval}$ **then**

**for** *each replica* **do**

1. Get $\Theta$ which features history of proposals $\Phi(\theta)$ and response $\lambda(L_{local})$

        2. Append proposal list to $X$

        3. Append likelihood list to $Y$

**end**

1. Train global surrogate model with input $X$ and output $Y$

      2. Save global surrogate model parameters

**end**

Increment $i$

**end**

**end**

---

### 3.3 Surrogate-assisted parallel tempering for Badlands model

The surrogate model learns from the relationship between the set of input parameters and response given by the true model. In our case, the input is the set of proposals giving by the sampler in the parallel tempering algorithm which features the proposals





for the Badlands model parameters such as rainfall and erodibility. The likelihood estimation by the surrogate model is called the pseudo-likelihood.

In a paralle computing environment we need to take into account the cost of inter-process communication which must be limited to avoid computational overhead. As given in our past implementation (Chandra et al., 2018c), the *swap interval* refers to the number of iterations after which each replica pauses and can undergo a replica transition. After the swap proposal is accepted or rejected, the replicas are resumed and they continue iterating while undergoing Metropolis transition in between the swap intervals. We note that the surrogate-assisted estimation is incorporated into the multi-core parallel tempering algorithm. In terms of the training procedure for the surrogate, (Chandra et al., 2018b) used a *surrogate interval* that determines the frequency of training by collecting the history of past samples with their likelihood from the respective replicas.

Taking into account that the true model is represented as $y = f(x)$, the surrogate model provides an approximation in the form $\hat{y} = \hat{f}(x)$, such that $y = \hat{y} + e$ where $e$ represents the difference or error. The task of the surrogate model is to provide pseudo-likelihood such that the the true likelihood is estimated by training from history of proposals which is given by the set of input $\mathbf{x_{r,s}}$ and likelihood $y_s$ where $s$ represents the sample and $r$ represents the replica. Hence, the training dataset $\Phi$ for the surrogate is developed by fusion of $\mathbf{x_{r,s}}$ across all the replica for a given surrogate interval $\psi$. Therefore, this can be formulated as follows.

$$
\begin{aligned}
\Phi &= (\mathbf{x_{1,s}}, \ldots, \mathbf{x_{1,s+\psi}}, \ldots, \mathbf{x_{M,s}}, \ldots, \mathbf{x_{M,s+\psi}}) \\
\lambda &= (y_{1,s}, \ldots, y_{1,s+\psi}, \ldots, y_{M,s}, \ldots, y_{M,s+\psi})
\end{aligned}
\tag{7}
$$

where $\mathbf{x_{r,s}}$ represents the set of parameters proposed at sample $s$, $y_{r,s} = \log\left(p(\mathbf{y}_{\mathcal{A}_{\mathbf{D},\mathbf{T}}}|\mathbf{x_{r,s}})\right)$ is the Gaussian likelihood which is dependent on data and the geo-scientific model, $M$ is the total number of replicas. The training surrogate dataset $\Theta$ features input $\Phi$ and response $\lambda$ at the end of every surrogate interval denoted by $s + \psi$. Hence, the pseudo likelihood $\hat{y}$ is given by $\hat{y} = \hat{f}(\Theta)$, where $\hat{f}$ is the surrogate model. The likelihood in training data is relaxed with respect of the temperature since it has been changed by taking $L_{local}/T_r$ for given replica $r$. We undo this change by multiplying the likelihood by the respective temperature which is a data processing step for surrogate model.

Algorithm 1 provides the details for execution of surrogate-assisted parallel tempering for the Badlands model. The algorithm begins by initializing the replicas that sample $\theta_n$ that represent Badlands model parameters such as rainfall and erodibility. This is done by drawing from a uniform distribution in a range $[\alpha, \alpha]$ where $\alpha$. The temperature ladder employs geometric ladder as given in Equation 4. Other key parameters include: 1. replica swap-interval $R_{swap}$, 2. maximum number of samples for each replica $R_{max}$, 3. surrogate interval, $S_{interval}$, and 4. surrogate probability $S_{prob}$. All of these values are determined experimentally.

After these are determined, the algorithm begins sampling for the respective replicas. Initially, the first surrogate interval considers the evaluation of all the proposals by the true likelihood function. Afterwards the data from the respective replicas are concatenated into training data $\Theta$ and used for training the surrogate model as shown in *State 31* of Algorithm 1. Once the surrogate is trained, it can be used to provide the pseudo-likelihood.



Given that the implementation considers each replica executed on a separate processing unit, a master processing unit is used to manage all the respective replicas as shown in Figure 4. The master process executes all the replicas in parallel and manages them by taking into account replica swap and surrogate training via the surrogate interval. The communication between the master process and the replica process requires inter-process communication protocols which is shown in Figure

4 and implemented by multi-processing libraries [1].

The pseudo-likelihood is utilized according to the *surrogate probability* as shown in State 6 in Algorithm 1. The surrogate model keeps updating its knowledge gained by data through observing the true likelihood from all of the replicas. The surrogate model is re-trained when remaining surrogate intervals are reached until the maximum sampling time is reached. At each every training interval, the surrogate model trains with knowledge from the previous state. Hence, the surrogate model remains up-

to-date and through transfer of knowledge form previous intervals, it gets better in estimation for the pseudo-likelihood. We note that only the samples associated with the true-likelihood becomes part of the surrogate training dataset.

Note that the training is done in the master process which features the global surrogate model as given in Figure 4. The replica processes provide the training dataset by file output which is read and concatenated by the master process. The way this is implemented is through having copies of the surrogate model (untrained one) in each of the replicas. After training, the

knowledge (i.e. weights of surrogate model) are transferred to each of the replicas as demonstrated in Figure 4. At the time of estimation for the pseudo-likelihood in each replica, we call the local surrogate that contains the knowledge from the global surrogate gained from the training data in the previous surrogate interval. The framework is flexible and hence the surrogate model at hand can be chosen by the user according to the nature of the likelihood surface.

The quality of estimation from the surrogate model can be validated by the root mean squared error (RMSE) which considers

the difference between the likelihood and the pseudo-likelihood. This can be seen as a regression problem with multi-input (parameters) and single output (likelihood). The RMSE is calculated by the following

$$RMSE = \sqrt{\frac{1}{N}\sum_{i=1}^{N}(y_i - \hat{y_i})^2}$$

where, $y_i$ and $\hat{y_i}$ are the true likelihood and the pseudo-likelihood value respectively and $N$ is the number of cases the surrogate is employed during sampling.

### 3.4   Surrogate model

The choice of the surrogate model needs to consider the computational resources taken for training the model during the sampling process. We note that Gaussian process, neural networks, and radial basis functions (Broomhead and Lowe, 1988), have been popular choices for surrogates in the literature.

In our case, we consider the inference problem that can feature, dozens, hundreds to thousands of parameters, hence the model needs to be efficiently trained without taking lots of computational resources. Moreover, the flexibility of the model to

have incremental training is also needed. Therefore, we rule out Gaussian process models since they have imitations in training

---

[1]We used Python multiprocessing library for implementation of multi-core parallel tempering: https://docs.python.org/2/library/multiprocessing.html



**Figure 4.** Surrogate-assisted parallel tempering framework. The training is done in the master process which features the global surrogate model. The replica processes provide the surrogate training dataset to the master process using inter-process communication. After training, the knowledge, i.e. weights of neural network based surrogate model, are transferred to each of the replicas.

given that the size of the dataset increases (Rasmussen, 2004). We use neural networks as the choice of the surrogate model in this study. The training data and neural network model can be formulated as follows.





The data given to the surrogate model is $\Phi$ and $\lambda$ as in Equation (7), where $\Phi$ is the input and $\lambda$ is the desired output of the model. The prediction of the model is denoted by $\hat{\lambda}$. We explain the surrogate models used in the paper as follows.

In our surrogate model, we consider a single hidden layer feedforward neural network as shown below. Given input $\mathbf{x_t}$, $f(\mathbf{x}_t)$ is computed by a feedforward neural network with one hidden layer defined by the function

$$f(\mathbf{x}_t) = g\left(\delta_o + \sum_{h=1}^{H} v_j g\left(\delta_h + \sum_{d=1}^{I} w_{dh}\mathbf{x_t}\right)\right) \tag{8}$$

where $\delta_o$ and $\delta_h$ are the bias weights for the output $o$ and hidden $h$ layer, respectively. $v_j$ is the weight which maps the hidden layer $h$ to the output layer. $w_{dh}$ is the weight which maps $\mathbf{x_t}$ to the hidden layer $h$ and $g$ is the activation function for the hidden and output layer units.

The only difference is that we use a different activation function $g(.)$ We use ReLU (rectified linear unitary function) as the activation function. The learning or optimization task then is to iteratively update the weights and biases to minimize the cross entropy loss $J(\mathbf{W}, \mathbf{b})$. This can be done using gradient update of weights using Adam (adaptive moment estimation) learning algorithm (Kingma and Ba, 2014) and stochastic gradient descent (Bottou, 1991, 2010). We experimentally evaluate them for training feedforward network for the surrogate model in the next section.

### 3.5 Design of Experiments

We provide an experimental study of the proposed surrogate-assisted parallel tempering (SAPT-Bayeslands) framework for selected landscape evolution problems. We compare the results with our parallel tempering Bayeslands framework (PT-Bayeslands) presented in an earlier study (Chandra et al., 2018c). The first part the experiments feature the accuracy of the surrogates in comparison with the actual model while the second part features the integration of SAPT for the Badlands model. We used *Keras* neural networks library (Chollet et al., 2015) for implementation of the surrogate. The open-source software package along with benchmark problems and experimental results is given here [2].

We first carry out an investigation of the effects of different surrogate training procedures and parameter evaluation for SAPT-Bayeslands using smaller problems. Afterwards, we apply the methodology to our selected landscape evolution problems. More specifically, the experiments are designed as follows.

- We generate a dataset for training and testing the surrogate for the Synthetic-Mountain and Continental-Margin landscape evolution problems. We use the neural network model for the surrogate and evaluate different training techniques.

- We evaluate if transfer of knowledge from previous surrogate interval is better than no transfer of knowledge for Synthetic-Mountain and Continental-Margin problems. Note this is done only with the data generated from previous step.

---

[2]Surrogate-assisted parallel tempering Bayeslands: https://github.com/badlands-model/surrogate-pt-Bayeslands



- We integrate the surrogate model into parallel tempering (SAPT-Bayeslands) and evaluate the effectiveness of the surrogate in terms of prediction of likelihood and overall time reduced is evaluated. Due to the requirement of extensive experimentation, only Synthetic-Mountain and Continental-Margin problems are considered.

- SAPT-Bayeslands is applied to the Tasmania landscape evolution problem and compared with PT-Bayeslands.

In SAPT-Bayeslands and PT-Bayeslands, we employ a random-walk proposal which is implemented by perturbing the chain in the respective replica with a small amount of Gaussian noise with a parameter specific step-size or standard deviation. The step-size $\beta_i$ for parameter $i$ is chosen to be a combination of a fixed step size $\phi = 0.02$, common to all parameters, multiplied by the range of possible values for parameter $i$ so that $\beta_i = (a_i - b_i) * \phi$, where, $a_i$ and $b_i$ represent the maximum and minimum limits of the priors for parameter and are given in Table 2.

Similarly, the geometric temperature ladder with maximum temperature $T_{max} = 10$ was used for determining the temperature level for for each of the replica. In trial experiments, the selection of these parameters depended on the accuracy. We used a replica exchange or swap interval, $R_{swap} = 10$ that determines when to check whether to swap with the neighboring replica. In previous work (Chandra et al., 2018c), it was observed that increasing the number of replicas up to a certain point does not necessarily mean that the computational time is lowered or better sampling is achieved. In this work, we limit the number of replicas as $R_{num} = 10$ for all experiments along with fixed maximum samples of 10 000 samples. We use a 15% burn-in which discards the portion of initial samples. This is a standard practice required for convergence which shows that the sampling discards the invariant and only considers the joint posterior distribution. The performance quality of the SAPT-Bayeslands and PT-Bayeslands framework is evaluated in terms of total simulation time, and root-mean-squared-error (RMSE) of the predicted elevation and erosion-deposition in the topography.

# 4 Results

## 4.1 Surrogate accuracy

In order to implement the surrogate model, we needs to evaluate the training algorithm such as Adam and stochastic gradient descent (SGD). Furthermore, we also evaluate certain parameters such as the size of the surrogate interval (batch-ratio), the neural network topology for the surrogate and the effectiveness of either training from scratch or to utilize previous knowledge for surrogate training (transfer and train). We create a training dataset from the cases where the true likelihood was used which compromises the history of the set of parameters proposed with the corresponding likelihood. This is done for standalone evaluation of the surrogate model which further ensures that the experiments are reproducible since different experimental runs will create different dataset depending on the exploration during sampling. Hence, we create a benchmark data set from history of samples proposed with their likelihood [3]. We then evaluate the neural network model designated for the surrogate using two major training algorithms which featured the Adam optimizer and stochastic gradient descent. The parameters that define the neural network surrogate model used for the experiments are given in Table 5. Note that the train size in Table 5 refers to the

---

[3] ,



maximum size of the data set. The training is done in batches where the batch ratio determines the training data set size as shown in Table 6.

**Table 5.** Neural network architecture for the different problems

| Dataset | Input | Output | Hidden layers [H1, H2, H3] | Train size | Test size |
|---|---|---|---|---|---|
| Continental-Margin | 6 | 1 | [64,35,24] | 8073 | 879 |
| Synthetic-Mountain | 5 | 1 | [65,35,25] | 8073 | 879 |

Table 6 presents the results for the experiments that took account of the training data collected during sampling for two benchmark problems (Continental-Margin and Synthetic-Mountain). The MSE indicates the performance of the surrogates

after the likelihood values (outcomes) in the dataset are normalized between [0,1]. Note that, we report the mean value of the mean-squared-error (MSE) for the given batch ratio from ten experiments. The batch ratio is taken in relation to the maximum number of samples across the chains ($R_{max}/R_{num}$). Although in most cases, the accuracy of the neural network is slightly better when training from scratch with combined data, howsoever, there is a huge trade off with the time required to train the network. The results show that the transfer and train methodology in general requires much lower computational time

when compared to training from scratch by combined data. Moreover, in comparison of SGD and Adam training algorithms, we observe that SGD achieves slightly better accuracy than Adam for Continental-Margin problem. However, Adam, having adaptive learning rate, outperforms SGD in terms of the time required to train the network. Thus, it can be summarized that transfer and train method is better since it saves significant computation time with a minor trade-off with accuracy.

### 4.2   Surrogate-assisted parallel tempering Bayeslands

In the experiments, we investigated the effects of the surrogate probability (s-prob) and surrogate interval (batch-ratio) on the the accuracy and time duration of the experiments. The accuracy of the prediction is evaluated by the mean square error (RMSE) of the predicted topography with the synthetic real topography, where elevation and erosion-deposition are reported. Note that the mean and standard deviation ( mean and std) of the accepted values of accuracy of prediction over the sampling is reported. The time is measured in seconds.

Table 7 and 8 shows the performance of the respective methods (PT-Bayeslands and SAPT-Bayeslands) with respective parameter settings for the Continental-Margin and Synthetic-Mountain problem. We observe that for the results regarding SAPT-Bayeslands, there not a significant difference in accuracy of elevation or erosion-deposition prediction given different values of surrogate probability. Howsoever, there is a significant difference in terms of the computational time saved. It is evident that greater surrogate probability gives more usage of surrogates through which more computational time is saved.

Furthermore, we notice that there is not a significant difference in accuracy of prediction or computational time given difference values of the batch-ratio. Figure 5 and 6 provides a visualization in the elevation prediction accuracy when compared to actual ground-truth between the two methods. Note that the prediction of erosion-deposition for 10 chosen points taken at selected locations shown in Table 4 is also given. Although both methods provide erosion-deposition prediction for 4 successive time



**Table 6.** Evaluation of surrogate training accuracy

| Dataset | Batch-ratio | Transfer and train | | | | Train from scratch | | | |
|---|---|---|---|---|---|---|---|---|---|
| | | SGD | | Adam | | SGD | | Adam | |
| | | MSE | Time(s) | MSE | Time(s) | MSE | Time(s) | MSE | Time(s) |
| Continental-Margin | 0.1 | 0.0198 | 19.40 | 0.0209 | 31.23 | 0.0199 | 88.17 | 0.0206 | 122.41 |
| | 0.2 | 0.0197 | 26.95 | 0.0211 | 56.84 | 0.0197 | 67.74 | 0.0199 | 100.49 |
| | 0.3 | 0.0199 | 25.53 | 0.0212 | 61.41 | 0.0197 | 70.71 | 0.0205 | 268.16 |
| | 0.4 | 0.0195 | 70.42 | 0.0193 | 48.28 | 0.0194 | 46.07 | 0.0188 | 140.90 |
| Synthetic-Mountain | 0.1 | 0.0161 | 40.38 | 0.0097 | 54.45 | 0.0161 | 282.0 | 0.0081 | 347.94 |
| | 0.2 | 0.0134 | 52.87 | 0.007 | 70.65 | 0.0139 | 185.025 | 0.007 | 857.38 |
| | 0.3 | 0.0129 | 65.105 | 0.0088 | 73.035 | 0.0123 | 179.36 | 0.0088 | 543.019 |
| | 0.4 | 0.0164 | 50.14 | 0.0048 | 87.67 | 0.0066 | 149.26 | 0.0038 | 653.85 |

intervals, we only show the final time interval due to lack of space for the respective problems. We notice that although the prediction accuracy is lower by SAPT-Bayeslands, the visualization shows that the mean prediction of the topography is close to ground-truth which is well covered by the credible interval. Figure 8 and Figure 9 show the the true likelihood and prediction by the surrogate for the Continental-Margin and Synthetic-Mountain problems, respectively. We notice that at certain intervals given in Figure 8, given by different replica, there is inconsistency in the predictions. Moreover, Figure 9 shows that the log-likelihood is very chaotic and hence there is difficulty in providing robust prediction at certain points in time given by samples for the respective replica.

Table 9 gives the results for Tasmania which is a bigger and computationally expensive problem. We select a good combination of the set of parameters evaluated in the previous experiments (s-prob = 0.5 and batch-ratio is 0.15). We used maximum of 10 000 samples with 10 replicas. We by notice that the performance of SAPT-Bayeslands is similar to PT-Bayeslands as shown in Figure 7 while 41.27 percentage of time is saved.

## 5 Discussion

We observe that the surrogate probability is directly related to the computational performance; this is obvious since computational time depends on how often the surrogate is used. Our concern is about the prediction performance especially while increasing the use of the surrogate as it could lower the accuracy which can results in poor estimation of the parameters. According to the results, the accuracy is well retained we give higher probability to the use of surrogates. In general, SAPT-Bayeslands achieves a lower prediction accuracy when compared to PT-Bayeslands. However, given the cross-section visualization, we find that the accuracy given in prediction by the surrogate based framework is not so poor. Moreover, application to a more computationally intensive problem (Tasmania) shows that a significant reduction in computational time is achieved.



**Table 7.** Surrogate evaluation for Continental-Margin problem

| Data-set | method | s-prob | batch-ratio | [Elevation] mean | std | [Erosion-Deposition] mean | std | time (min) | time saved (%) |
|---|---|---|---|---|---|---|---|---|---|
| Continental-Margin | PT-Bayeslands | N/A | N/A | 60.05 | 10.45 | 49.23 | 14.65 | 84.50 | N/A |
| | SAPT-Bayeslands | 0.25 | 0.10 | 119.37 | 31.48 | 106.13 | 32.54 | 78.36 | 7.27 % |
| | SAPT-Bayeslands | 0.25 | 0.15 | 138.41 | 22.14 | 124.30 | 29.24 | 74.98 | 11.27 % |
| | SAPT-Bayeslands | 0.25 | 0.20 | 123.09 | 37.00 | 112.45 | 35.45 | 76.77 | 9.15 % |
| | SAPT-Bayeslands | 0.50 | 0.10 | 137.86 | 29.42 | 123.89 | 27.87 | 49.89 | 40.96 % |
| | SAPT-Bayeslands | 0.50 | 0.15 | 131.14 | 37.31 | 117.59 | 34.58 | 54.27 | 35.78 % |
| | SAPT-Bayeslands | 0.50 | 0.20 | 130.74 | 36.59 | 120.30 | 30.34 | 56.46 | 33.18 % |
| | SAPT-Bayeslands | 0.75 | 0.10 | 126.16 | 29.50 | 116.11 | 26.23 | 34.17 | 65.48 % |
| | SAPT-Bayeslands | 0.75 | 0.15 | 127.60 | 32.73 | 115.08 | 34.48 | 34.32 | 59.38 % |
| | SAPT-Bayeslands | 0.75 | 0.20 | 125.18 | 33.70 | 114.73 | 37.86 | 36.98 | 56.24 % |

**Table 8.** Surrogate evaluation results for Synthetic-Mountain. Mean Squared Error (MSE) values and Time elapsed for various surrogate intervals and probabilities

| Data-set | method | s-prob | batch-ratio | Elevation mean | std | Erosion-Deposition mean | std | time (min) | time saved (%) |
|---|---|---|---|---|---|---|---|---|---|
| Synthetic-Mountain | PT-Bayeslands | N/A | N/A | 4.87 | 1.68 | 1.41 | 0.34 | 128.20 | N/A |
| | SAPT-Bayeslands | 0.25 | 0.10 | 17.51 | 32.05 | 5.09 | 12.32 | 100.77 | 21.40 % |
| | SAPT-Bayeslands | 0.25 | 0.15 | 22.50 | 28.90 | 7.97 | 12.16 | 101.98 | 20.45 % |
| | SAPT-Bayeslands | 0.25 | 0.20 | 11.66 | 26.65 | 3.11 | 10.38 | 110.57 | 13.75 % |
| | SAPT-Bayeslands | 0.50 | 0.10 | 18.79 | 35.75 | 5.51 | 14.11 | 71.35 | 44.34 % |
| | SAPT-Bayeslands | 0.50 | 0.15 | 23.67 | 30.34 | 8.59 | 12.83 | 75.21 | 41.33 % |
| | SAPT-Bayeslands | 0.50 | 0.20 | 12.77 | 28.95 | 3.61 | 11.42 | 80.33 | 37.34 % |
| | SAPT-Bayeslands | 0.75 | 0.10 | 26.99 | 42.75 | 8.69 | 17.06 | 44.72 | 65.12 % |
| | SAPT-Bayeslands | 0.75 | 0.15 | 24.18 | 30.31 | 8.75 | 12.66 | 49.64 | 61.28 % |
| | SAPT-Bayeslands | 0.75 | 0.20 | 11.49 | 25.63 | 2.89 | 9.33 | 54.91 | 57.17 % |

**Table 9.** Surrogate evaluation for Continental-Margin problem

| Data-set | method | s-prob | batch-ratio | [Elevation] mean | std | [Erosion-Deposition] mean | std | time (min) | time saved (%) |
|---|---|---|---|---|---|---|---|---|---|
| Tasmania | PT-Bayeslands | N/A | N/A | 197.27 | 23.42 | 3.9 | 0.5 | 4724.47 | N/A |
| | SAPT-Bayeslands | 0.50 | 0.20 | 235.79 | 32.06 | 3.91 | 0.1 | 2774.53 | 41.27 % |







(a) Continental Margin (PT-Bayeslands)

(b) Continental Margin (SAPT-Bayeslands)

(c) Continental Margin (PT-Bayeslands)

(d) Continental Margin (SAPT-Bayeslands)

**Figure 5.** Cross section of prediction for Continental-Margin problem. The prediction of erosion-deposition for 10 chosen points in the topography is also given.



(a) Synthetic-Mountain (PT-Bayeslands)

(b) Synthetic-Mountain (SAPT-Bayeslands)

(c) Synthetic-Mountain (PT-Bayeslands)

(d) Synthetic-Mountain (SAPT-Bayeslands)

**Figure 6.** Cross section of prediction for Synthetic-Mountain problem. The prediction of erosion-deposition for 10 chosen points in the topography is also given.




(a) Tasmania (PT-Bayeslands)

(b) Tasmania (SAPT-Bayeslands)

(c) Tasmania (PT-Bayeslands)

(d) Tasmania (SAPT-Bayeslands)

**Figure 7.** Cross section of prediction for Tasmania problem along with the prediction of erosion-deposition for 10 chosen points.





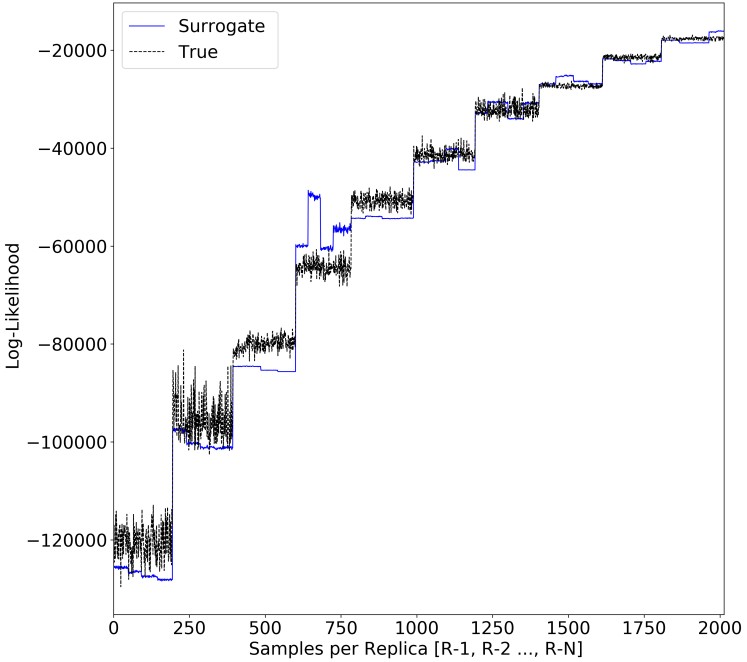

**Figure 8.** Surrogate likelihood vs true likelihood estimation for Continental-Margin topography

The initial evaluation for the setup surrogate model shows that its is best to use a transfer learning approach where the knowledge from the past surrogate interval is utilized and refined with new surrogate data. This consumes much less time than accumulating data and training the surrogate from scratch at every surrogate interval. We note that in cases when the surrogate model is used, there is no prediction given by the model. Hence, the predictions (elevation and erosion-deposition) during

5 sampling are gathered only from the true Badlands model evaluation rather than the surrogate. In this way, one could argue that the surrogate model is not mimicking the true model; however, we are guiding the sampling algorithm towards forming better proposals without evaluation of the true model. A direction forward is in incorporating other forms of surrogates which could be in terms of running low resolution Badlands model as the surrogate which would be computationally faster in evaluating the proposals. Furthermore, computationally efficient implementations of landscape evolution models that only feature landscape

10 evolution (Braun and Willett, 2013) could be used as the surrogate while Badlands that features both landscape evolution and erosion-deposition formation could be used as the true model. Computationally efficient implementations of landscape evolution models that consider parallel processing (Hassan et al., 2018) could also be used in the Bayeslands framework. In this case, the challenge would be in allocating special processing cores for Badlands and others for parallel tempering.





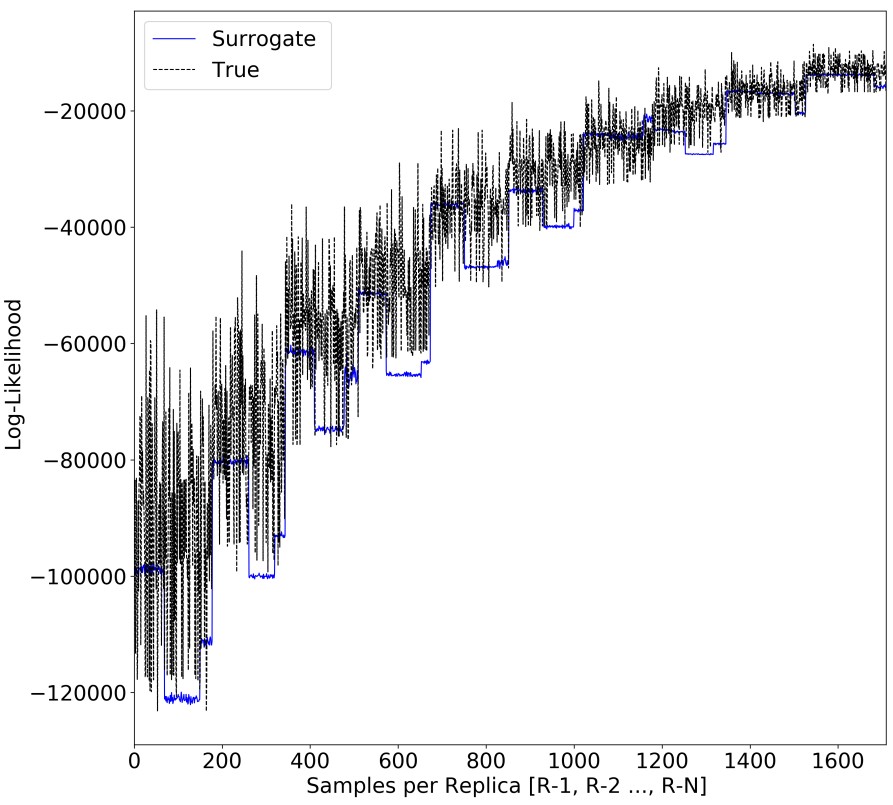

**Figure 9.** Surrogate likelihood vs true likelihood estimation for Synthetic-Mountain topography

The surrogate framework was adapted from (Chandra et al., 2018b) with major difference of featuring gradient-based proposals. Gradient-based learning or parameter estimation has been very popular in machine learning due to availability of gradient information. Due to the complexity in geological or geophysical numerical forward models, it is difficult to obtain gradients which has been the case of Badlands landscape evolution model. We use random-walk proposals which is a canonical sampling approach with a number of limitations. Hence, we need to incorporate advanced meta-heuristic techniques to form non-gradient based proposals for efficient search. Our study is limited to a fairly small seat of free parameters and a major challenge would be to develop surrogate models with an increased set of parameters.

## 6 Conclusions

We presented a novel application of surrogate-assisted parallel tempering that features parallel computing for landscape evolution models using Badlands. Initially, we experimented with two different approaches for training the surrogate model where we found that transfer learning based approach is beneficial and could help reduce computational time of the surrogate. Using this approach, we present the experiments that featured evaluating certain key parameters of the surrogate-based framework. In general, we observe that the proposed framework lowers computational time significantly, while maintaining the required quality in parameter estimation and uncertainty quantification.

In future work, we envision to apply the proposed framework to more complex applications such as evolution of continental-scale landscapes and basins over millions of years. The approach could be used for other forward models such as those that feature geological reef development or lithospheric deformation. Furthermore, the posterior distribution of our parameters require multi-modal sampling methods; hence a combination of meta-heuristics for proposals with surrogate assisted parallel tempering could improve exploration features and also help in lowering the computational costs.

*Code availability.* https://github.com/intelligentEarth/surrogate-pt-Bayeslands

*Author contributions.* R. Chandra led the project and contributed in writing the paper and designing the experiments. D.Azam contributed by running experiments and providing documentation of the results. A. Kapoor contributed in programming, running experiments and providing documentation of the results. D. Müller contributed by managing the project, writing the paper and providing analysis of the results.

*Competing interests.* No competing interests are present.

*Acknowledgements.* We would like to thank Konark Jain for technical support.



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
