# Peer review of "Surrogate-assisted Bayesian inversion for landscape and basin evolution models"

_Geoscientific Model Development, 2018_

## Referee Comment (RC1) · Anonymous Referee #1 · 19 Jun 2019

Description

This paper attempts to present Bayesian inference for an inverse problem in landscape and sedimentary basin evolution models, with acceleration by parallel tempering and use of a surrogate likelihood in the form of a neural net that is incrementally trained along the chain using evaluations of the true model. The paper attempts optimization to perform inversion, and the chain adapts the surrogate.

Comments on content

I will restrict my comments to the computational Bayesian aspects as I am no expert in models for landscape and sedimentary basin evolution. However, the paper makes virtually no mention of the evolution model used, and the majority concerns Bayesian computation, so my comments actually account for the brunt of the content of this paper.

Parallel tempering is an arcane algorithm that probably gives no advantage

One can see from the references that the computational Bayesian method used in this paper predates 1997, which is when the first serious attempts at Bayesian computation for inverse problems were made. There has been tremendous development in algorithms since then, particularly in terms of efficiency with respect to inverse problems. This manuscript seems entirely unaware of these developments. (The more recent statistical references in this manuscript are not to sampling methods, and frequently to the authors' own papers.)

It was recognized soon after 1997 that while parallel tempering is a nice idea, it is not very effective for inverse problems, and suffers from fundamental difficulties. One such major difficulty is the need for tuning "psuedo priors" to allow the parallel chains to mix – this is mentioned in the original paper [MP92] (cited in the present manuscript) and studied in subsequent papers, but no mention of this issue is made in the present manuscript. It also became known that parallel tempering's use of parallel resources is essentially trivial; one can see from the distribution for the ensemble (Eqn. on p 4 line 3) that the parallel chains are statistically independent, so there is no efficiency gain over simply running a parallel instance of single chains that move up and down tempering levels. That multilevel algorithm has been implemented in multiple guises, most effectively in the delayed acceptance algorithm [CF05] and multilevel versions,

and the variance-reduction methods of multi-level Monte Carlo [DKST15].

These algorithms significantly outperform parallel tempering as used in this manuscript; see [HRMVF11] for a review of some of these topics. On the other hand, there is no evidence presented in the present manuscript that parallel tempering actually leads to improved computational efficiency (beyond running separate parallel chains), and my impression is that it does not

A further note is that the present manuscript makes no mention of the random-walk proposal used, though this is VERY critical to the efficiency of the method. There are modern methods that use *correlated* parallel chains and use the information across chains to adapt to an optimal random-walk proposal [SOLHTJ12] that would be much more efficient than the method used in this manuscript.

**0.0.1 The proposed method is not ergodic for the target distribution**

One can see in Alg. 1 (p11) that the surrogate is trained, or *adapted* in the language of MCMC, as the algorithm proceeds. Hence this MCMC is not stationary and does not satisfy the conditions for standard MCMC to be ergodic for the desired target distribution. Indeed, it is easy to see from the structure of Alg. 1 that it will not target the desired posterior distribution. That is to say, Alg. 1 converges (if it converges at all) to some distribution that is not the target. So it is somehow useless for performing a quantitative solution of this inverse problem.

The present literature contains potential fixes to these problems, yet the present manuscript makes no mention of them. In particular, the surrogate transition method [L01] would allow Alg. 1 to correctly use a fixed surrogate, *at no increase in computational cost*, while the adaptive algorithm in [CFO11] gives a framework for provably ergodic methods that accommodate the adaptation of the surrogate, again *at no increase in computational cost.*

[Figure]

**0.0.2  The algorithm in the manuscript is not run to convergence**

This is evident from Fig. 8, and elsewhere.

**0.0.3  This manuscript does not implement Bayesian methods**

Bayesian analysis produces a posterior distribution over possible solutions to the inverse problem. It is then necessary to summarize the posterior distribution, in a way that is appropriate for the problem at hand. Without specific evidence, there is no reason that optimizing the posterior distribution to give the MAP estimate, as attempted in this manuscript, is a good summary statistic. Indeed, it is known that in even moderate dimension inverse problems the MAP estimate can be arbitrarily far from the bulk of feasible solutions, and can be sensitive to noise realizations – that is to say that it is a hopeless summary statistic. The computation attempted in this manuscript is not a sensible summary statistic for a Bayesian analysis.

Also, the use of Bayesian modelling in this manuscript is entirely bogus. For example, the sum-of-squares log likelihood (p10 l 6, and elsewhere) has no physical meaning for sediment transport, while plenty of more physically-realistic measures are available. Thus, the likelihood function used is not a sensible Bayesian model. Prior models are implicitly uniform, which makes no sense in terms of Bayesian probabilistic modelling of the model parameters. Uncertainty in distributions, via hyperpriors, is not even considered.

This manuscript widely advertises its Bayesian credentials and the use of Bayesian inference (in the title, abstract, introduction), yet does not implement any sensible Bayesian methods. It certainly does not achieve "a rigorous approach to uncertainty quantification" (p2, l1).

**0.0.4 The technical writing is poor in places**

For example, lines 5, 6, 7 on p10 notation is inconsistent: "The likelihood function $L_e(\theta)$ is given by $L_l(\theta) = ...$ where the subscript $e$, in $L_e(\theta)$," then again $L_l(\theta)$ used in l 19 p10, and so on. Line 9-10 in Alg. 1, $L_{local}$ is not defined. Line 5 in Alg. 1, proposal density is not defined. Line 22 on p1 "deterministic geophysical forward model can be seen as a probabilistic model via Bayesian inference" is nonsense.

**0.0.5 There are some positive aspects about this manuscript that could be publishable**

Some of the computed results in Fig. 2, 5, 6 look interesting, to my untrained eye. The idea of using a neural net surrogate to accelerate computation is interesting, though a somewhat obvious one given the current hype around neural nets. Nevertheless, the attempt to use neural nets in this way is interesting. I must reemphasize, as stated above, that the use of the surrogate ought to be performed within one of the well-established algorithms for correct use of a surrogate, while the algorithm in this manuscript is ad-hoc and incorrect. The analysis in this manuscript is actually a maximum likelihood calculation – though these estimates are known to suffer from quantitative problems, as outlined above, and notwithstanding the issue with the unphysical likelihood modelling, mentioned above. The authors might consider re-presenting this work with an accurate description of the calculation as a maximum likelihood (and dropping all mention of Bayesian methods, which appears to be far from the authors' skill set).

Conclusions

This manuscript makes no contemporary contribution to Bayesian methods or computation. Indeed, in terms of Bayesian computation, the methods presented in this manuscript constitute a step backwards by some decades, while attention to convergence is completely absent and hence a major deficiency. The probabilistic modelling is crude, to the point of being worthless. The use of parallel chains and a neural net surrogate seem more an exercise in programming than an efficient, quantified solution to a scientific problem. Publishing this paper would be a disservice to the community as it grossly misrepresents the current literature on Bayesian modelling and computation for this problem. The current literature contains better and more computationally efficient solutions.

If this manuscript were submitted to a journal that deals with Bayesian computation, I doubt the editor would even send it out for review.

I recommend this manuscript be rejected.

**References**

Marinari, E. and Parisi, G.: Simulated tempering: a new Monte Carlo scheme, EPL (Europhysics Letters), 19, 451, 1992.

Christen, J.A. and Fox, C., 2005. Markov chain Monte Carlo using an approximation. Journal of Computational and Graphical statistics, 14(4), pp.795-810.

Dodwell, T.J., Ketelsen, C., Scheichl, R., Teckentrup, A.L.: A hierarchical multilevel Markov chain Monte Carlo algorithm with applications to uncertainty quantification in subsurface flow. SIAM/ASA Journal on Uncertainty Quantification 3(1), 1075–1108 (2015)

Higdon, D., Reese, C.S., Moulton, J.D., Vrugt, J.A. and Fox, C., 2011. Posterior exploration for computationally intensive forward models. Handbook of Markov Chain Monte Carlo, pp.401-418.

Solonen, A., Ollinaho, P., Laine, M., Haario, H., Tamminen, J. and Järvinen, H., 2012. Efficient MCMC for climate model parameter estimation: Parallel adaptive chains and early rejection. Bayesian Analysis, 7(3), pp.715-736.

Liu, J.S., 2001. Monte Carlo strategies in scientific computing. Springer Series in Statistics.

Cui, T., Fox, C. and O'Sullivan, M.J., 2011. Bayesian calibration of a large-scale geothermal reservoir model by a new adaptive delayed acceptance Metropolis Hastings algorithm. Water Resources Research, 47(10).

---

## Author Comment (AC1) · 19 Jun 2019

It seems that the reviewer has has failed to understand that parallel tempering MCMC is a methodology for implementing Bayesian inference. The reviewer needs to review MCMC methods and will find that parallel tempering is a well established method that has been widely used for inversion problems in Earth science. Furthermore, it is widely used in areas of physics, such as astrophysics, environmental modelling and many other areas that have inversion problems. A simple search in google scholar will help.

One of the key researchers in this area is Malcolm Sambridge from ANU who has more recently used parallel tempering MCMC (Bayesian method) for inversion problems in Earth Sciences!

[Figure]

Malcolm Sambridge, A Parallel Tempering algorithm for probabilistic sampling and multimodal optimization Geophysical Journal International, Volume 196, Issue 1, January, 2014, Pages 357–374, https://doi.org/10.1093/gji/ggt342

Dosso SE1, Holland CW, Sambridge M., Parallel tempering for strongly nonlinear geoacoustic inversion. J Acoust Soc Am. 2012 Nov;132(5):3030-40. doi: 10.1121/1.4757639.

M. Sambridge, "A Parallel Tempering algorithm for probabilistic sampling and multimodal optimization," in Geophysical Journal International, vol. 196, no. 1, pp. 357-374, Jan. 2014. doi: 10.1093/gji/ggt342 URL: http://ieeexplore.ieee.org/stamp/stamp.jsp?tp=&arnumber=8179653&isnumber=8179647

Furthermore, the reviewer has undermined the novelty of this work, which is particularly intended for computationally expensive model optimisation (inference). The contribution of the paper is not in improving parallel tempering MCMC but in applying it for landscape evolution problems and further to use surrogate assisted methods for further improving computational efficiency.

Latest MCMC methods such as Hamiltonian MCMC and Langevin based MCMC methods cannot be applied to this problem since there is no gradient information. This is major reason parallel tempering MCMC with random-walk proposals has been used.

The paper not only formulates the concept, but also implements and releases open source software that can help the landscape evolution modelling committee.

We note that there are related journals that could publish the paper with but the authors found this journal to be most suitable. The authors strongly rejects the biased review comments of the reviewer that shows lack of understanding in basic concepts in Bayesian inference. Hence, the reviewers calls for review process to consider those who know about MCMC methods with applications to Earth science models!
* * *
[Figure]

2019.

---

## Referee Comment (RC2) · Anonymous Referee #2 · 19 Nov 2019

The scope of this study is quite large, with many complex details included. For example, Bayesian inversion is described with on a complex model, an MCMC implementation is used and outlined which involves Parallel Tempering and also the parallel architecture of the authors code is also described in some detail. These are all relevant, if not original features of the author's computer code, and it seems there is an intent is to describe them all. While completeness is a good thing, and the intent is appreciated here, it can tend to obscure the main focus which is to evaluate the effectiveness of combining surrogate 'pseudo-likelihoods' which are trained 'in-situ' during the sampling process. The authors may improve the effectiveness of the manuscript by describing some, or most, of these algorithmic details in appendices, using the main text to give an overview and clearer focus on the primary point, i.e. the comparison between surrogate

and full forward modelling in the Bayesian sampling.

Having said that I think the authors should be commended for attempting to include full detail, which is appreciated. Since the primary focus here (should be) on the surrogate model, I found the section which described this, and both approaches for training, rather light. I didn't get a clear enough understanding to give me confidence that I could reproduce it. Given the central importance of this aspect I suggest an appendix devoted to describing the structure of the Neural network and its training be described carefully in an appendix. That is outside of the context it is being used, i.e. without reference to MCMC or PT or even Figure 4. The readers should have a clear picture of this as a 'stand alone' component of the overall algorithm.

All figures and tables need much better captions. They appear to be an afterthought. Variables, axes and details of the figure need to be explained or define. I suggest even including a hint at what you want the reader to notice in each figure/table. At present they are just titles.

Section 2.1 needs to be re-examined. $E(x)$ is not defined, nor is $W\_L$ . Presumably eqn. 920 is the MCMC balance condition. If so where is the prior ratio, where is the proposal ratio. If it is assumed that these cancel this needs to be specified and explained.

The use of terms local and global need clarification, as far as I can tell it refers to things that happen on a parallel compute node compared to the master. Correct? Please explain. Not clear the distinction needs to be made.

There a reasonably large number of choices that need to be made for control parameters throughout, intervals of PT and surrogate, sizes of training sets, starting value sof Neural network etc. Several of which are listed on page 12 lines 25-30. We are told that 'All of these values are determined experimentally', but how? While I trust that the author has done a competent job, we still need to know what criteria were used to decide between desirable and less desirable values? Some explanation is required.

There are numerous typographical and grammatical errors throughout the manuscript which creates a poor impression. Below are a few I identified, but there are sufficiently regular to warrant a careful proofread by an independent person in any future version. I suspect this has not happened prior to submission. In some parts the text descends into obscure technical detail regarding data flow in the parallel structure, etc. Again independent feedback from a colleague might sort these issues out.

The repeated use of the term 'replica' is confusing. This appears to be describing unrelated models (sets of variables) at the time step of an McMC chain. In what sense are they replicas? My understanding is that the only thing in general any two 'replica's have in common is the same chain index.

I was unsure what the actual numbers of unknowns and what the typical compute cost of a Likelihood evaluation were in each experiment. It would be best to explicitly state this in each case, as it puts the calculations and MCMC sampling into perspective. I did not get a clear picture of this.

In page 10 line 20, it may be useful to mention that these are what is known as hierarchical MCMC models, as the variance of both data types are being treated as unknowns. This is not a good aspect I think but one that is glossed over.

The bottom line message from this manuscript as I understood it was that across several examples shown, both the computationally inexpensive 'Continental-Margin' and 'Synthetic-Mountain' cases as well as the more computationally demanding 'Tasmania' case there are time savings of between 7 and 65% when using the surrogate over the full forward model. Necessarily these numbers depend on details of tuning various control parameters and other choices made, and I assume a good job has been done. However whether this is of practical significance is not clear. If I had a computer that was three times as fast as the one used here then presumably I would achieve the same compute time as the surrogate with the more accurate full physics based model. Correct? While I think a saving has been demonstrated, the author should

really comment on the significance of the observed improvement in compute time.

As the author clearly points out well, the improved efficiency of the surrogate-assisted MCMC sampler comes at the cost of lower accuracy as measured ultimately in the Bayesian mean and standard deviations of the Elevation and Erosion-Deposition parameters. As I understood it the PT-Bayeslands results are considered the 'near truth' and the Surrogate-assisted, SAPT-Bayesland, as the approximate. So perhaps the more important question, is then is how to judge whether the trade-off of accuracy against compute time is significant. One way this might be done is ask whether the PT-Bayeslands could produce the same if not better accuracy than SAPT-Bayesland with the same computation budget, i.e. fewer samples. I assume it is possible to do such an experiment by rescaling the number of samples available to PT-Bayeslands by the relative compute times observed in the experiments. This question/experiment has not been addressed but it would be instructive to try it. Again the central question is one of significance of the results. It would be impressive for the reader to see some attempt along these lines.

Overall I think this is an encouraging piece of work which could be significantly improved by a restructured manuscript and more quantitative evaluation on the two points above.

Some typo and grammatical errors: P1 L10: 'has been with successfully' - ? P8 Figure 3b is missing? P11 Last sentence starting 'In our case,...' contains 'giving by the sampler' meaning? I did not actually understand this sentence at all. P12 L3 'paralle' P12 L12: 'the the true Likelihood' P12 L18 Is this the Gaussian Likelihood or the log-Likelihood? P13 L30 'they have imitations in training' ? P15 eqn (8) balance size of brackets. P15 L11 define 'J(W,b)' P16 L11 'for for' P16 L22 'we needs' P16 L29 where is footnote 3? P17 L23 'Howsoever' ? P18 L10 'We by notice that' P18 L13 what is surrogate probability? Do you mean accuracy in recovering marginal probability?

---

## Author Comment (AC2) · 25 Nov 2019

The authors sincerely thank the reviewer for these comments. In the revision, we have created an Appendix section that features the details of parallel tempering MCMC and Training algorithms for neural network surrogate model.

We are including python code with the paper along with data and sample results in order to ensure reproducibility. We also revised the Algorithm 1 to ensure that we make the method clearer and have amended the texts in these sections (highlighted in light brown) to ensure that all the details are presented clearly.

Comment: The use of terms local and global need clarification, as far as I can tell it refers to things that happen on a parallel compute node compared to the master.

Correct? Please explain. Not clear the distinction needs to be made.

Response: This has been added: "In surrogate-assisted parallel tempering, global surrogate essentially refers to the main surrogate model that features training data combined from different replicas running in parallel cores. Local surrogate model refers to the surrogate model in the given replica that incorporate knowledge from the global surrogate in order to make a prediction given new input data (sample of proposal). Note that the training only takes place in the global surrogate and the prediction or estimation for pseudo-likelihood only takes place in the local surrogates. "

Comment: However whether this is of practical significance is not clear. If I had a computer that was three times as fast as the one used here then presumably I would achieve the same compute time as the surrogate with the more accurate full physics based model. Correct? While I think a saving has been demonstrated, the author should really comment on the significance of the observed improvement in compute time. As the author clearly points out well, the improved efficiency of the surrogate-assisted MCMC sampler comes at the cost of lower accuracy as measured ultimately in the Bayesian mean and standard deviations of the Elevation and Erosion-Deposition parameters.

Response: The following has been added in the discussion section: " In general, the proposed method achieves a lower prediction accuracy when compared to PT-Bayeslands. However, given the cross-section visualization, we find that the accuracy given in prediction by the surrogate based framework is not so poor. Moreover, application to a more computationally intensive problem (Tasmania) shows that a significant reduction in computational time is achieved. We demonstrated the method using small models that run in seconds or minutes, Computational costs of continental scale Badlands models is very large (5 kilometer resolution for Australian continent for 149 million years is about 72 hours) and hence, in case when thousands of samples need to be drawn, the use of surrogates can be very useful. However, we note that improved efficiency of the surrogate-assisted Bayeslands comes at the cost of lower accuracy and

there is a trade-off between accuracy and computational time."

Comment: I assume it is possible to do such an experiment by rescaling the number of samples available to PT-Bayeslands by the relative compute times observed in the experiments. This question/experiment has not been addressed but it would be instructive to try it. Again the central question is one of significance of the results. It would be impressive for the reader to see some attempt along these lines.

Response: The results in Table 7 and 8 show computation time reduced and RMSE accuracy, with a fixed number of samples to provide a fair comparison. In our previous work, we have already shown the performance trend of PT-Bayeslands given different number of samples (Chandra et. a, 2019) https://agupubs.onlinelibrary.wiley.com/doi/abs/10.1029/2019GC008465

Furthermore, the following has been added in the discussion: "The results in terms of prediction accuracy given by the proposed method can be further improved in future work with the way the surrogate is trained. Rather than a global surrogate model, local surrogate model on its own can be used, where the training only takes place in the local surrogates by only relying on history of the likelihood and hence taking a univariate time series prediction approach using neural networks. Our major contribution is in terms of the parallel computing based open-source software and the proposed underlying framework for incorporating surrogates, taking into account complex issues such as inter-process communication. This opens the road to try different types of surrogate models while using the underlying framework and open source software. "

---

## Author Response (AR2)

Description

 This paper attempts to present Bayesian inference for an inverse problem in landscape and sedimentary basin evolution models, with acceleration by parallel tempering and use of a surrogate likelihood in the form of a neural net that is incrementally trained along the chain using evaluations of the true model. The paper attempts optimization to perform inversion, and the chain adapts the surrogate. The authors sincerely thank the reviewer for time and comments.

Comments on content I will restrict my comments to the computational Bayesian aspects as I am no expert in models for landscape and sedimentary basin evolution. However, the paper makes virtually no mention of the evolution model used, and the majority concerns Bayesian computation, so my comments actually account for the brunt of the content of this paper.  Please note that in the revised paper, Section 2.2 gives an overview of the Badlands landscape evolution model. Section 3.1 gives problem definition for the Badlands model along with the respective topography datasets.

Parallel tempering is an arcane algorithm that probably gives no advantage One can see from the references that the computational Bayesian method used in this paper predates 1997, which is when the first serious attempts at Bayesian computation for inverse problems were made. There has been tremendous development in algorithms since then, particularly in terms of efficiency with respect to inverse problems.This manuscript seems entirely unaware of these developments. (The more recent statistical references in this manuscript are not to sampling methods, and frequently to the authors' own papers.) It was recognized soon after 1997 that while parallel tempering is a nice idea, it is not very effective for inverse problems, and suffers from fundamental difficulties. One such major difficulty is the need for tuning "psuedo priors" to allow the parallel chains to mix – this is mentioned in the original paper [MP92] (cited in the present manuscript) and studied in subsequent papers, but no mention of this issue is made in the present manuscript. It also became known that parallel tempering's use of parallel resources is essentially trivial; one can see from the distribution for the ensemble (Eqn. on p 4 line 3) that the parallel chains are statistically independent, so there is no efficiency gain over simply running a parallel instance of single chains that move up and down tempering levels. That multilevel algorithm has been implemented in multiple guises, most effectively in the delayed acceptance algorithm [CF05] and multilevel versions, and the variance-reduction methods of multi-level Monte Carlo [DKST15]. These algorithms significantly outperform parallel tempering as used in this manuscript; see [HRMVF11] for a review of some of these topics.

 There is significant work done in the area of parallel tempering MCMC with applications to Astrophysics and Geoscience problems in the past two decades (after 1997). Parallel tempering is a well established MCMC method that has been widely used for inversion problems in Earth science. Furthermore, it is widely used in areas of physics, such as astrophysics, environmental modelling and many other areas that have inversion problems.   One of the key researchers in this area is Malcolm Sambridge from ANU who has more recently used parallel tempering MCMC (Bayesian method) for inversion problems in Earth Sciences and some of his prominent works are as follows that has attracted a wide range of citations and shared the geo-scientific modelling community:

1. Malcolm Sambridge, A Parallel Tempering algorithm for probabilistic sampling and multimodal optimization Geophysical Journal International, Volume 196, Issue 1, January, 2014, Pages 357–374, https://doi.org/10.1093/gji/ggt342
2. Dosso SE1, Holland CW, Sambridge M., Parallel tempering for strongly nonlinear geoacoustic inversion. J Acoust Soc Am. 2012 Nov;132(5):3030-40. doi: 10.1121/1.4757639.
3. M. Sambridge, "A Parallel Tempering algorithm for probabilistic sampling and multimodal optimization," in Geophysical Journal International, vol. 196, no. 1, pp. 357-374, Jan. 2014. doi: 10.1093/gji/ggt342 URL: http://ieeexplore.ieee.org/stamp/stamp.jsp?tp=&arnumber=8179653&isnumber=8179647

On the other hand, there is no evidence presented in the present manuscript that parallel tempering actually leads to improved computational efficiency (beyond running separate parallel chains), and my impression is that it does not A further note is that the present manuscript makes no mention of the random-walk proposal used, though this is VERY critical to the efficiency of the method. There are modern methods that use correlated parallel chains and use the information across chains to adapt to an optimal random-walk proposal [SOLHTJ12] that would be much more efficient than the method used in this manuscript. 0.0.1 The proposed method is not ergodic for the target distribution One can see in Alg. 1 (p11) that the surrogate is trained, or adapted in the language of MCMC, as the algorithm proceeds. Hence this MCMC is not stationary and does not satisfy the conditions for standard MCMC to be ergodic for the desired target distribution. Indeed, it is easy to see from the structure of Alg. 1 that it will not target the desired posterior distribution.

In the revision of the paper, we have mentioned that we are using Random-walk and Adaptive Random walk proposed, given their details in Section 3.5 of the manuscript.

We note that the proposed surrogate framework is general and any other MCMC algorithms can be used besides parallel tempering MCMC, given that they can be parallelised.

That is to say, Alg. 1 converges (if it converges at all) to some distribution that is not the target. So it is somehow useless for performing a quantitative solution of this inverse problem. The present literature contains potential fixes to these problems, yet the present manuscript makes no mention of them. In particular, the surrogate transition method [L01] would allow Alg. 1 to correctly use a fixed surrogate, at no increase in computational cost, while the adaptive algorithm in [CFO11] gives a framework for provably ergodic methods that accommodate the adaptation of the surrogate, again at no increase in computational cost
0.0.2 The algorithm in the manuscript is not run to convergence This is evident from Fig. 8, and elsewhere. 0.0.3

The revised manuscript has added Section 4.2 that gives results for convergence diagnosis with different proposal distributions and methods.

This manuscript does not implement Bayesian methods Bayesian analysis produces a posterior distribution over possible solutions to the inverse problem. It is then necessary to summarize the posterior distribution, in a way that is appropriate for the problem at hand. Without specific evidence, there is no reason that optimizing the posterior distribution to give the MAP estimate, as attempted in this manuscript, is a good summary statistic. Indeed, it is known that in even moderate dimension inverse problems the MAP estimate can be arbitrarily far from the bulk of feasible solutions, and can

be sensitive to noise realizations – that is to say that it is a hopeless summary statistic. We are not giving a MAP estimate directly, but integrating out the posterior distribution using Bayesian inference MCMC.

The computation attempted in this manuscript is not a sensible summary statistic for a Bayesian analysis. Also, the use of Bayesian modelling in this manuscript is entirely bogus. For example, the sum-of-squares log likelihood (p10 l 6, and elsewhere) has no physical meaning for sediment transport, while plenty of more physically-realistic measures are available. The contribution of this paper is not about data sources but the use of surrogates to assist sampling procedure for computationally expensive models. Our previous paper has already addressed the issues regarding the likelihood and data: (Chandra et. a, 2019)
https://agupubs.onlinelibrary.wiley.com/doi/abs/10.1029/2019GC008465

Thus, the likelihood function used is not a sensible Bayesian model. Prior models are implicitly uniform, which makes no sense in terms of Bayesian probabilistic modelling of the model parameters. Uncertainty in distributions, via hyperpriors, is not even considered. This manuscript widely advertises its Bayesian credentials and the use of Bayesian inference (in the title, abstract, introduction), yet does not implement any sensible Bayesian methods. It certainly does not achieve "a rigorous approach to uncertainty quantification" (p2, l1). We have limited or sparse data and it's difficult to gather informative priors. The approach is taking into account the Bayesian methodology, we have priors, it's just that they are less informative, but we do account for rigorous   uncertainty quantification since we are integrating out the posterior using MCMC.

For example, lines 5, 6, 7 on p10 notation is inconsistent: "The likelihood function $L_e(\theta)$ is given by $L_l(\theta) = ...$ where the subscript e, in $L_e(\theta)$," then again $L_l(\theta)$ used in l 19 p10, and so on. Line 9-10 in Alg. 1, Llocal is not defined. Line 5 in Alg. 1, proposal density is not defined. Line 22 on p1 "deterministic geophysical forward model can be seen as a probabilistic model via Bayesian inference" is nonsense. 0.0.5
 Our previously published papers contribution was the likelihood function and the Bayesian model: (Chandra et. a, 2019) https://agupubs.onlinelibrary.wiley.com/doi/abs/10.1029/2019GC008465 whereas the focus of this paper is the surrogate model to enhance computation. There are many geophysical papers that essentially show: "deterministic geophysical forward model can be seen as a probabilistic model via Bayesian inference", such as those by Malcom Sambrige as mentioned earlier.

There are **some positive aspects about this manuscript** that could be publishable. Some of the computed results in Fig. 2, 5, 6 look interesting, to my untrained eye. **The idea of using a neural net surrogate to accelerate computation is interesting**, though a somewhat obvious one given the current hype around neural nets. The authors sincerely thank the reviewer for these positive comments.

 Nevertheless, the attempt to use neural nets in this way is interesting. I must reemphasize, as stated above, that the use of the surrogate ought to be performed within one of the well-established algorithms for correct use of a surrogate, while the algorithm in this manuscript is ad-hoc and incorrect. The analysis in this manuscript is actually a maximum likelihood calculation – though these estimates are known to suffer from quantitative problems, as outlined above, and notwithstanding the issue with the unphysical likelihood modelling, mentioned above. The authors might consider

re-presenting this work with an accurate description of the calculation as a maximum likelihood (and dropping all mention of Bayesian methods, which appears to be far from the authors' skill set)

The major contribution of this paper which is about enhancing existing work with use of surrogates. The proposed framework can be used for other MCMC methods.

Conclusions This manuscript makes no contemporary contribution to Bayesian methods or computation. Indeed, in terms of Bayesian computation, the methods presented in this manuscript constitute a step backwards by some decades, while attention to convergence is completely absent and hence a major deficiency.

The probabilistic modelling is crude, to the point of being worthless. The use of parallel chains and a neural net surrogate seem more an exercise in programming than an efficient, quantified solution to a scientific problem. Publishing this paper would be a disservice to the community as it grossly misrepresents the current literature on Bayesian modelling and computation for this problem.
The current literature contains better and more computationally efficient solutions. If this manuscript were submitted to a journal that deals with Bayesian computation, I doubt the editor would even send it out for review. I recommend this manuscript be rejected.
There is a multidisciplinary focus of the paper. The above comments are too focused on validating Bayesian inference methodology, whereas the paper is about using existing Bayesian inference methodology and applying for Geoscientific models such as landscape evolution models. The contribution of the paper is in the area of Geoscientific models, not in the area of Bayesian computation. The use of parallel tempering MCMC and contribution to its theoretical foundations are still active in statistics and machine learning journals:

1. Desjardins, G., Courville, A., Bengio, Y., Vincent, P., & Delalleau, O. (2010, May). Parallel tempering for training of restricted Boltzmann machines. In *Proceedings of the thirteenth international conference on artificial intelligence and statistics* (pp. 145-152). Cambridge, MA: MIT Press.
2. Sambridge, M. (2013). A parallel tempering algorithm for probabilistic sampling and multimodal optimization. *Geophysical Journal International*, *196*(1), 357-374.
3. Miasojedow, B., Moulines, E., & Vihola, M. (2013). An adaptive parallel tempering algorithm. *Journal of Computational and Graphical Statistics*, 22(3), 649-664.
4. Baragatti, M., Grimaud, A., & Pommeret, D. (2013). Likelihood-free parallel tempering. *Statistics and Computing*, *23*(4), 535-549.
5. Li, Y., Protopopescu, V. A., Arnold, N., Zhang, X., & Gorin, A. (2009). Hybrid parallel tempering and simulated annealing method. *Applied Mathematics and Computation*, *212*(1), 216-228.
6. Katzgraber, H. G., Trebst, S., Huse, D. A., & Troyer, M. (2006). Feedback-optimized parallel tempering Monte Carlo. *Journal of Statistical Mechanics: Theory and Experiment*, *2006*(03), P03018.

References Marinari, E. and Parisi, G.: Simulated tempering: a new Monte Carlo scheme, EPL (Europhysics Letters), 19, 451, 1992. Christen, J.A. and Fox, C., 2005. Markov chain Monte Carlo using an approximation. Journal of Computational and Graphical statistics, 14(4), pp.795-810. Dodwell, T.J., Ketelsen, C., Scheichl, R., Teckentrup, A.L.: A hierarchical multilevel Markov chain Monte Carlo algorithm with applications to uncertainty quantification in subsurface flow. SIAM/ASA

Journal on Uncertainty Quantification 3(1), 1075–1108 (2015) Higdon, D., Reese, C.S., Moulton, J.D., Vrugt, J.A. and Fox, C., 2011. Posterior exploration for computationally intensive forward models. Handbook of Markov Chain Monte Carlo, pp.401- 418.

Solonen, A., Ollinaho, P., Laine, M., Haario, H., Tamminen, J. and Järvinen, H., 2012. Efficient MCMC for climate model parameter estimation: Parallel adaptive chains and early rejection. Bayesian Analysis, 7(3), pp.715-736. Liu, J.S., 2001. Monte Carlo strategies in scientific computing. Springer Series in Statistics. Cui, T., Fox, C. and O'Sullivan, M.J., 2011. Bayesian calibration of a large-scale geothermal reservoir model by a new adaptive delayed acceptance Metropolis Hastings algorithm. Water Resources Research, 47(10).

The novelty of this work is for computationally expensive model inference. The contribution of the paper is not in improving parallel tempering MCMC but in applying it for landscape evolution problems and further to use surrogate assisted methods for further improving computational efficiency. Latest MCMC methods such as Hamiltonian MCMC and Langevin based MCMC methods cannot be applied to this problem since there is no gradient information. This is a major reason parallel tempering MCMC with random-walk  and adaptive random-walk proposals has been used. The paper not only formulates the concept, but also implements and releases open source software that can help the landscape evolution modelling committee.

The authors sincerely thank the reviewer for valuable time and comments.

**Anonymous Referee #2**

The scope of this study is quite large, with many complex details included. For example, Bayesian inversion is described with on a complex model, an MCMC implementation is used and outlined which involves Parallel Tempering and also the parallel architecture of the authors code is also described in some detail. These are all relevant, if not original features of the author's computer code, and it seems there is an intent is to describe them all. While completeness is a good thing, and the intent is appreciated here, it can tend to obscure the main focus which is to evaluate the effectiveness of combining surrogate 'pseudo-likelihoods' which are trained 'in-situ' during the sampling process. The authors may improve the effectiveness of the manuscript by describing some, or most, of these algorithmic details in appendices, using the main text to give an overview and clearer focus on the primary point, i.e. the comparison between surrogate and full forward modelling in the Bayesian sampling. Having said that I think the authors should be commended for attempting to include full detail, which is appreciated. The authors sincerely thank the reviewer for these comments. In the revision, we have created Appendix section that features the details of parallel tempering MCMC and Training algorithms for neural network surrogate model.

Since the primary focus here (should be) on the surrogate model, I found the section which described this, and both approaches for training, rather light. I didn't get a clear enough understanding to give me confidence that I could reproduce it.
We are including python code with the paper along with data and sample results in order to ensure reproducibility. We also revised the Algorithm 1 to ensure that we make the method clearer and have amended the texts in these section (highlighted in light brown) to ensure that all the details are presented clearly.

Given the central importance of this aspect I suggest an appendix devoted to describing the structure of the Neural network and its training be described carefully in an appendix. In the revision, the authors have added the basic neural network model and details about the training equations in the Appendix.

That is outside of the context it is being used, i.e. without reference to MCMC or PT or even Figure 4. The readers should have a clear picture of this as a 'stand alone' component of the overall algorithm. All figures and tables need much better captions. The caption has been improved in this figure and further references are given in the text. Other captions have also been edited/extended.

They appear to be an afterthought. Variables, axes and details of the figure need to be explained or define. I suggest even including a hint at what you want the reader to notice in each figure/table. At present they are just titles. The variables and axes in all the figures have been defined - in the captions.

Section 2.1 needs to be re-examined. E(x) is not defined, nor is W_L . Presumably eqn. 920 is the MCMC balance condition. This paragraph has been moved to applentix where L(x) is used instead of E(x) to define the likelihood function. W_L is changed to L_local which refers to the likelihood ratio for the proposals.and

If so where is the prior ratio, where is the proposal ratio. If it is assumed that these cancel this needs to be specified and explained. The prior ratio cancels out since we use uniform priors and has been stated in the text.

The use of terms local and global need clarification, as far as I can tell it refers to things that happen on a parallel compute node compared to the master. Correct? Please explain. Not clear the distinction needs to be made.

This has been added:
"In surrogate-assisted parallel tempering, global surrogate essentially refers to the main surrogate model that features training data combined from different replicas running in parallel cores. Local surrogate model refers to the surrogate model in the given replica that incorporates knowledge from the global surrogate in order to make a prediction given new input data (sample of proposal). Note that the training only takes place in the global surrogate and the prediction or estimation for pseudo-likelihood only takes place in the local surrogates. "

There a reasonably large number of choices that need to be made for control parameters throughout, intervals of PT and surrogate, sizes of training sets, starting values of Neural network etc. Several of which are listed on page 12 lines 25-30. We are told that 'All of these values are determined experimentally', but how? While I trust that the author has done a competent job, we still need to know what criteria were used to decide between desirable and less desirable values? Some explanation is required.  "In trial experiments, the selection of these parameters depended on the performance in terms of the number of accepted samples and prediction accuracy of elevation and sediment/deposition. "

There are numerous typographical and grammatical errors throughout the manuscript which creates a poor impression. Below are a few I identified, but there are sufficiently regular to warrant a careful proofread by an independent person in any future version. I suspect this has not happened prior to submission. These are fixed in the revision.

In some parts the text descends into obscure technical detail regarding data flow in the parallel structure, etc. Again independent feedback from a colleague might sort these issues out. The repeated use of the term 'replica' is confusing. This appears to be describing unrelated models (sets of variables) at the time step of an McMC chain. In what sense are they replicas? Replicas have different temperature levels as defined by the temperature ladder.
My understanding is that the only thing in general any two 'replica's have in common is the same chain index. Yes
I was unsure what the actual numbers of unknowns and what the typical compute cost of a Likelihood evaluation were in each experiment. It would be best to explicitly state this in each case, as it puts the calculations and MCMC sampling into perspective. I did not get a clear picture of this. Time taken to run a single model is  shown in Table 1

In page 10 line 20, it may be useful to mention that these are what is known as hierarchical MCMC models, as the variance of both data types are being treated as unknowns. This is not a good aspect I think but one that is glossed over. We have two components of the likelihood, the elevation and sediment deposition which are evaluated jointly. We have added this sentence to clarify further:

"We note that given that the sediment erosion/deposition is temporal, we could have a hierarchical Bayesian model (Chib and Carlin,1999; Wikle et al.,1998) with two stages for MCMC sampling, that evaluates the respective likelihoods, which could be future work. "

The bottom line message from this manuscript as I understood it was that across several examples shown, both the computationally inexpensive 'Continental-Margin' and 'Synthetic-Mountain' cases as well as the more computationally demanding 'Tasmania' case there are time savings of between 7 and 65% when using the surrogate over the full forward model. Necessarily these numbers depend on details of tuning various control parameters and other choices made, and I assume a good job has been done. Thanks for these comments. We ran many trial experiments to fine tune parameters and also create synthetic problems that had different levels of computational time to demonstrate the method.

However whether this is of practical significance is not clear. If I had a computer that was three times as fast as the one used here then presumably I would achieve the same compute time as the surrogate with the more accurate full physics based model. Correct? While I think a saving has been demonstrated, the author should really comment on the significance of the observed improvement in compute time. As the author clearly points out well, the improved efficiency of the surrogate-assisted MCMC sampler comes at the cost of lower accuracy as measured ultimately in the Bayesian mean and standard deviations of the Elevation and Erosion-Deposition parameters.

Actually, we had a major bug in the code and had to carry out most of the experiments again. Now our results show that the surrogate Bayeslands approach gives similar predictions as canonical Bayeslands, but reduces the computational time significantly (see Figure 5 and Figure 6 and Table 8).

The following has been added in the discussion section:

 "Given the cross-section presented in the results for Continental-Margin and Synthetic Mountain problems, we find that there is not much difference in the accuracy given in prediction by the SAPT-Bayeslands when compared to PT-Bayeslands. Moreover, the application to a more computationally intensive problem (Tasmania), we find that a significant reduction in computational time is achieved. Although we demonstrated the method using small-scale  models that run within a few seconds to minutes,  the computational costs of continental-scale Badlands models is extensive. For instance, the computational time for a  5-kilometre resolution for Australian continent Badlands model for 149 million years is about 72 hours, and hence, in the case when thousands of samples are required,  the use of surrogates can be beneficial. We note that improved efficiency of the surrogate-assisted Bayeslands comes at the cost of  accuracy for some problems (in case of Tasmania problem), and there is a trade-off between accuracy and computational time."

 As I understood it the PT-Bayeslands results are considered the 'near truth' and the Surrogate-assisted, SAPT-Bayesland, as the approximate. So perhaps the more important question, is then is how to judge whether the trade-off of accuracy against compute time is significant. One way this might be done is ask whether the PT-Bayeslands could produce the same if not better accuracy than SAPT-Bayesland with the same computation budget, i.e. fewer samples. Actually, this is the

main goal of our experiments and we have already demonstrated it in the results. It seems  we have not been very clear and in the revision we are further highlighting the goal of our experiments.

 I assume it is possible to do such an experiment by rescaling the number of samples available to PT-Bayeslands by the relative compute times observed in the experiments. This question/experiment has not been addressed but it would be instructive to try it. Again the central question is one of significance of the results. It would be impressive for the reader to see some attempt along these lines.

The results in Table 7  show computation time reduced and RMSE accuracy, with a fixed number of samples to provide a fair comparison.  In our previous work, we have already shown the performance trend of PT-Bayeslands given different number of samples (Chandra et. a, 2019) https://agupubs.onlinelibrary.wiley.com/doi/abs/10.1029/2019GC008465

Furthermore, the following has been added in the discussion:

"The results in terms of prediction accuracy  given by the proposed method can be further improved in future work with  the way the surrogate is trained. Rather than a global surrogate model, local surrogate model on its own can be used, where the training only takes place in the local surrogates by only relying on history of the likelihood and hence taking a univariate  time series prediction approach using neural networks. Our major  contribution is in terms of the parallel computing based open-source software and the proposed underlying framework for incorporating surrogates, taking into account complex issues such as inter-process communication. This opens the road to try different types of surrogate models while using the underlying framework and open source software. "

Overall I think this is an encouraging piece of work which could be significantly improved by a restructured manuscript and more quantitative evaluation on the two points above. The authors sincerely thank the reviewer for valuable time and comments.

Some typo and grammatical errors: P1 L10: 'has been with successfully' - ? P8 Figure 3b is missing? Fixed

P11 Last sentence starting 'In our case,. . .' contains 'giving by the sampler' meaning? I did not actually understand this sentence at all.  The sentence has been revised: " In  our case, the input is the set of proposals by the respective replica  samplers in the parallel tempering algorithm."

P12 L3 'paralle' P12 L12: 'the the true Likelihood'  fixed

P12 L18 Is this the Gaussian Likelihood or the logLikelihood?  Log-likelihood - fixed

P13 L30 'they have imitations in training' ?  fixed

P15 eqn (8) balance size of brackets.  fixed

P15 L11 define 'J(W,b)' P16 L11 'for for'  fixed fixed

P16 L22 'we needs' P16 L29 where is footnote 3?  fixed

P17 L23 'Howsoever' ? P18 L10 'We by notice that'  fixed

P18 L13 what is surrogate probability? -
 Do you mean accuracy in recovering marginal probability? " ----determines the frequency of employing the surrogate model for estimating the pseudo-likelihood." has been added in the text.

The authors sincerely thank the reviewer for valuable time and comments.